# Mediolateral foot placement control can be trained: Older adults learn to walk more stable, when ankle moments are constrained

**Mohammadreza Mahaki**[1,2‡], **Anina Moira van Leeuwen**[1,2,3‡]*, **Sjoerd M. Bruijn**[1,2,3], **Nathalie van der Velde**[2,4], **Jaap H. van Dieën**[1,2]

**1** Department of Human Movement Sciences, Faculty of Behavioural and Movement Sciences, Amsterdam Movement Sciences, Vrije Universiteit Amsterdam, Amsterdam, The Netherlands, **2** Amsterdam Movement Sciences, Research Program(s), Amsterdam, The Netherlands, **3** Institute of Brain and Behavior, Amsterdam, The Netherlands, **4** Department of Internal Medicine/Geriatrics, Amsterdam University Medical Centers, University of Amsterdam, Amsterdam, The Netherlands

‡ MM and AML Shared first authors on this work.
* moiravleeuwen@gmail.com

**Data Availability Statement:** The relevant data are available from Zenodo at DOI:10.5281/zenodo. 8368670 [https://zenodo.org/record/8368670].

## Abstract

Falls are a problem, especially for older adults. Placing our feet accurately relative to the center-of-mass helps us to prevent falling during gait. The degree of foot placement control with respect to the center-of mass kinematic state is decreased in older as compared to young adults. Here, we attempted to train mediolateral foot placement control in healthy older adults. Ten older adults trained by walking on shoes with a narrow ridge underneath (LesSchuh), restricting mediolateral center-of-pressure shifts. As a training effect, we expected improved foot placement control during normal walking. A training session consisted of a normal walking condition, followed by a training condition on LesSchuh and finally an after-effect condition. Participants performed six of such training sessions, spread across three weeks. As a control, before the first training session, we included two similar sessions, but on normal shoes only. We evaluated whether a training effect was observed across sessions and weeks in a repeated-measures design. Whilst walking with LesSchuh, the magnitude of foot placement error reduced half-a-millimeter between sessions within a week (cohen's $d$ = 0.394). As a training effect in normal walking, the magnitude of foot placement errors was significantly lower compared to the control week, by one millimeter in weeks 2 (cohen's $d$ = 0.686) and 3 (cohen's $d$ = 0.780) and by two millimeters in week 4 (cohen's $d$ = 0.875). Local dynamic stability of normal walking also improved significantly. More precise foot placement may thus have led to improved stability. It remains to be determined whether the training effects were the result of walking on LesSchuh or from repeated treadmill walking itself. Moreover, enhancement of mechanisms beyond the scope of our outcome measures may have improved stability. At the retention test, gait stability returned to similar levels as in the control week. Yet, a reduction in foot placement error persisted.

**Funding:** This study was funded by the 2020 Amsterdam Movement Sciences talent grant, obtained by M. Mahaki and A.M. van Leeuwen. In addition S.M. Bruijn and A.M. van Leeuwen were funded by a grant from the Netherlands Organization for Scientific Research (016. Vidi.178.014), https://www.nwo.nl/en/.

**Competing interests:** The authors have declared that no competing interests exists.

## Introduction

Anyone who sees someone close to them grow older may become concerned about them falling. Indeed, older adults are less stable than young adults [1–3] and at old age a fall can have severe consequences [4]. Falls in older adults most commonly occur during walking [5], suggesting that improving gait stability may prevent falls. In young adults, foot placement control with respect to variations in the center-of-mass (CoM) kinematic state is the dominant mechanism to maintain gait stability [6–11]. One of the reasons why older adults are at a higher risk of falling may be a compromised control over foot placement [12].

Foot placement control is achieved through modulation of hip muscle activity [8, 13], based on a combination of visual, vestibular and proprioceptive information [12, 14–16]. This allows adequate foot placement in relation to the CoM kinematic state. Older adults demonstrated less well coordinated foot placement, due to impaired (processing of) proprioceptive information, or inability to generate adequate motor responses [12, 17]. Perhaps, training older adults, by imposing constraints demanding more accurate foot placement, will help them to relearn how to coordinate their foot placement with respect to the CoM.

The relative variance explained ($R^2$) by a model predicting foot placement based on the CoM kinematic state [7, 11] describes what proportion of the variance in foot placement is explained by the variance in CoM kinematic state. It can thus be interpreted as a measure quantifying how well foot placement is coordinated with respect to the CoM and we have coined it "the degree of foot placement control" [18].

Constraining ankle moments by walking on the so-called LesSchuh, a shoe with a narrow ridge along the length of the sole, led to an increase in this degree of degree of foot placement control in young adults [19]. The reduced ability to correct for foot placement errors, due to constrained ankle moment control [20], may have driven this adaptation, and this suggests a training potential. Still so far, we have been unsuccessful to transfer this adaptation to normal walking (without ankle moment constraints). Despite a trend, no significant after-effect was found in young adults after training with LesSchuh [21]. It must be noted, however, that in the latter study participants walked on a split-belt treadmill, for which the effect of ankle moment constraints on foot placement control is different [19]. Apart from treadmill interaction effects, the already high degree of foot placement control in young adults may have prevented a training effect. Alternatively, it may require multiple training sessions before a significant improvement in the degree of foot placement control can be detected.

In this study, we investigated whether foot placement control can be improved in older adults, by walking with constrained ankle moments (i.e. by walking on LesSchuh). We expected a greater training potential in older adults due to their lower degree of foot placement control [12]. We asked them to train over a training period of three weeks, with two sessions per week. Each week, the ridge of the LesSchuh on which they walked became narrower. We hypothesized that the degree of foot placement control would improve between sessions and weeks. In addition to the degree of foot placement control (relative explained variance), we considered the magnitude of foot placement errors. We also calculated local divergence exponents [22], step width and stride time, to explore changes in stability (control).

## Methods

Ten older adults ($\geq$ 65 years old, 7 males & 3 females) participated in this repeated-measures study. All participants filled out an inclusion/exclusion questionnaire (see S4 File), before they were allowed to participate. If the questionnaire raised any doubt for inclusion, we discussed this with the participant, in the end ensuring that all participants included could sustain the intensity of the training protocol. All participants gave written informed consent, and ethical

approval for this experiment had been granted by the ethical committee of the faculty of behavioral and movement science of the Vrije Universiteit Amsterdam (VCWE-2020-186R1, January fifth 2021). We started recruiting participants in April 2021, either in the mall, or through word of mouth. All participants completed eight sessions, spread across one control and three training weeks. Eight participants completed the ninth session, the retention test, within the second week after the last training week. Two participants performed the retention test respectively four and nine weeks after the last training week, due to their holidays. Data were collected during all sessions. Data was stored by participant code, and thus anonymously. However, there is a key linking the participant's name to their participant code, which was only accessed by the primary researchers (M. Mahaki and A.M. van Leeuwen). This key was used to ensure the data of each session was attributed to the corresponding participant, as well as to compile a personal report for each participant, after analyzing the results and writing of the manuscript. Each participant received their personal report for personal use.

The data and code for analysis can be found at: https://doi.org/10.5281/zenodo.8368670.

## Study design

All participants followed the training protocol as presented in Fig 1. The training protocol comprised four weeks, of which the first week served as a control week. In week one, participants only walked on normal shoes. In weeks two to four, participants still walked on normal shoes in the normal walking and after-effect conditions. In addition, participants trained by walking on the treadmill with LesSchuh in the training conditions (Fig 2). As such, the sessions in week two until four were designed similarly as the single training session of our previous study [21]. LesSchuh is a shoe that has a narrow ridge underneath the sole, which allows for anteroposterior roll-off and push-off, but limits mediolateral center of pressure shifts. As such, it constrains mediolateral ankle moment control. The width of the ridge was narrowed every training week (weeks two until four) from two centimeter to respectively one-and-a-half and finally one centimeter. If, with the narrower ridge, the participant could no longer perform the training according to our instructions, we allowed them to walk on the ridge with the same width as the preceding week.

Each (training) session consisted of five bouts of treadmill walking, intermitted with opportunities to rest. The first bout (normal walking condition) lasted ten minutes, and bouts two to five lasted five minutes each. Between the last two bouts no rest was offered. In this intermission, we quickly changed the shoes back to normal shoes, to be able to evaluate immediate after-effects (see S2 File).

To conclude the experiment, we conducted a retention test. During the retention test participants walked for five minutes with normal shoes, followed by five minutes on LesSchuh with the same width of the ridge as in their final training session.

## General assessment

During the first session, participants were first asked to fill out the Mini Mental State Examination (MMSE), to validate inclusion (MMSE>24). Furthermore, we performed the Short Physical Performance Battery (SPPB) [23], to assess the overall fitness of the older adults. Finally, we assessed their concern of falling by the Falls Efficacy Scale–International (FES-I) [24].

## Equipment and safety precautions

To measure the kinematics of the feet and the thorax (as a proxy for the center of mass), we tracked three cluster markers (with three single markers each) with two Optotrak cameras sampled at 100 Hz (Northern Digital Inc, Waterloo Ont, Canada). Participants walked at a

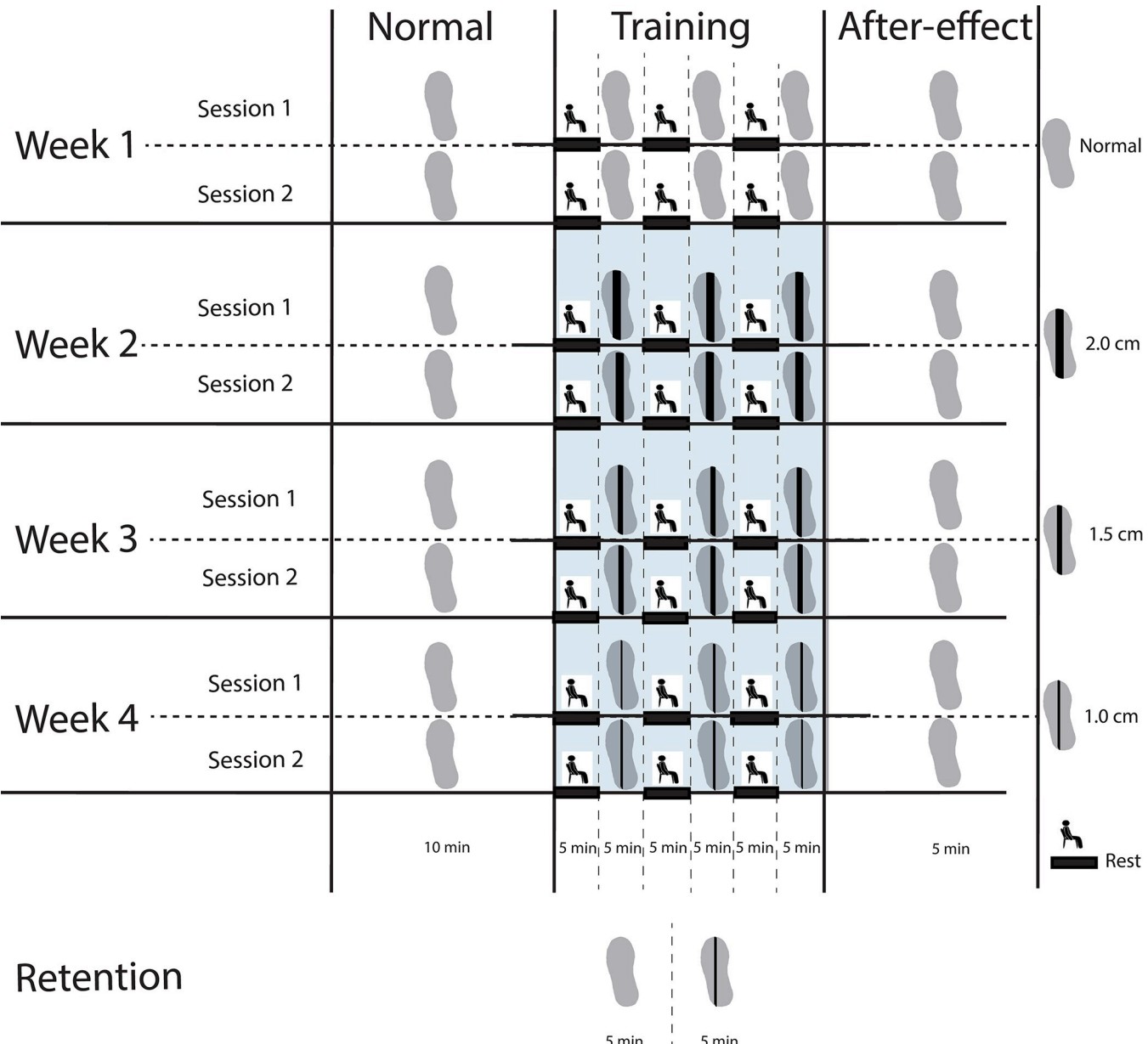

**Fig 1. Experimental design.** Each week consisted of two (training) sessions. In week 1, participants walked only on normal shoes. In weeks 2–4, they walked on normal shoes in the Normal walking and After-effect conditions, but on LesSchuh (Fig 2) in the Training condition (shaded blue area). In weeks 2–4, the widths of the ridges underneath the shoes were 2.0, 1.5 and 1.0 cm respectively. In week 6, we performed a retention test. We collected data during all sessions.

three-meter long treadmill, wearing a safety harness connected to the ceiling to support the participant in case of a fall (Fig 3).

## Experimental protocol

We asked participants to wear shoes to which the cluster markers were attached. We invited participants to stand on the treadmill, where we fastened the safety harness. We also attached an elastic band to mount the thorax cluster markers. In the first session, we determined the participant's preferred walking speed on normal shoes, by increasing and decreasing the speed

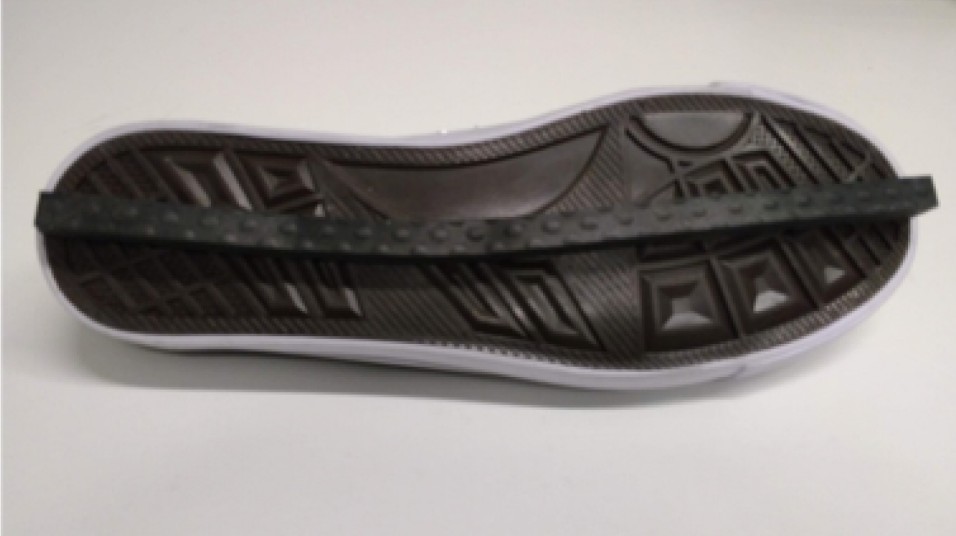

**Fig 2. LesSchuh.** The narrow ridge underneath the sole constrains mediolateral shifts in the center of pressure. The width of the ridge in the figure is one centimeter.

until the participant felt comfortable with the pace. This walking speed was used in all subsequent conditions, sessions and weeks, to ensure that any changes in foot placement control could be attributed to constraining ankle moment control, rather than to changes in gait speed [25].

While walking on LesSchuh, participants were instructed to only stand on the narrow ridge underneath the sole. Furthermore, we asked them to keep pointing their feet straight ahead (avoiding a toeing-out strategy [26]). If corrections were needed, we gave feedback (e.g. "rotate your left foot more inward", "keep pointing your feet straight ahead", "stay on the ridge") and we kept motivating participants throughout the trial. If it was evident that the participant had a hard time adhering to the instructions, we tried not to push them further than trying their best. Whenever feedback distracted the participant, we limited our feedback.

### Clinical outcome measures and questionnaire

As clinical outcome measures, baseline SPPB and FES-I scores were obtained from the general assessment at the first experimental session. The walking part of the SPPB was repeated at the end of weeks one, two and three. At the end of week four (after the last training session), the full SPPB and FES-I were re-assessed. We also administered an additional questionnaire, asking participants about their (training) experiences and whether they felt the training had any positive impact (see S3 File). Finally, at the start of the retention test we repeated the SPPB and FES-I.

### Biomechanical outcome measures

Our main outcome measure was "the degree of foot placement control" as assessed by the relative explained variance ($R^2$) of Model (1), in which FP represents the demeaned mediolateral placement of the swing foot relative to the stance foot (step width), $CoM_{pos}$ the demeaned mediolateral CoM position and $CoM_{vel}$ the demeaned mediolateral CoM velocity at terminal swing. $\beta_{pos}$, $\beta_{vel}$ and $\varepsilon$ represent respectively the regression coefficients of $CoM_{pos}$ and $CoM_{vel}$, and the residual (i.e. the foot placement error). FP was determined as the mediolateral distance between the heel markers at midstance. $CoM_{pos}$ and $CoM_{vel}$ were determined with respect to

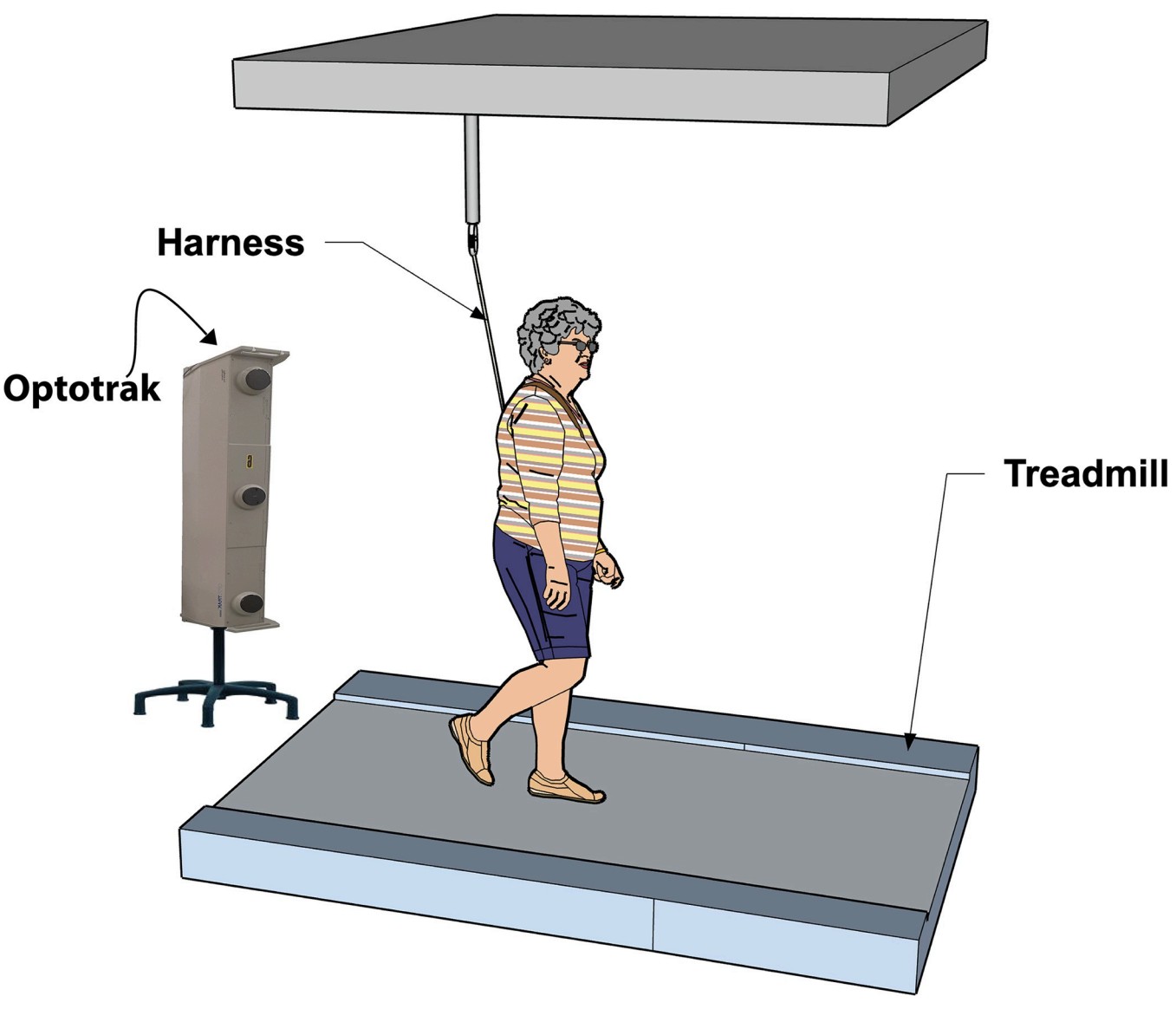

**Fig 3. Treadmill with safety harness.**

the stance foot.

$$FP = \beta_{pos} \cdot CoM_{pos} + \beta_{vel} \cdot CoM_{vel} + \varepsilon, \qquad [1]$$

Complementary to the degree of foot placement control ($R^2$) we calculated "the magnitude of foot placement error", as the standard deviation of the residual ($\varepsilon$) of Model (1). In our previous work, we referred to this measure as "precision in foot placement control" [21]. The degree of foot placement control describes what percentage of foot placement variance can be explained by variations in the CoM kinematic state during the preceding swing phase. However, as it is a relative measure, the $R^2$'s value can be in- or deflated depending on the total variance. It thus does not reflect how precise foot placement is in absolute terms. By considering the magnitude of the foot placement error as an extra outcome measure, we add an absolute measure of foot placement precision.

As secondary outcome variables we evaluated step width, stride time (time between two subsequent heelstrikes of the same leg), and local dynamic stability. For the latter, we computed the local divergence exponent [22, 27–29] of the mediolateral CoM velocity. First, we time-normalized the signal, so that on average each stride was 100 samples in length. Then we constructed a six-dimensional state space based on copies with a 25-samples time delay. Within this state space, for each point, we found the five nearest neighbors (defined as the points that had the smallest Euclidian distance to the original point, while having at least half an average stride time of temporal separation). Subsequently, the divergence with the nearest neighbors was tracked for 1000 samples. We fitted a line (least squares fit) through the first 50 samples (i.e. half a stride on average) of the averaged logarithmic divergence curve. The slope of this line defined the local divergence exponent. The lower the local divergence exponent, the more stable the gait pattern.

The outcome measures as described above were analyzed in a similar way as in [21]. Accordingly, we distinguished between "normal" (first 10 minutes of each session), "training" (concatenated 3 x 5 minutes walking on LesSchuh of each session) and "after-effect" (last 5 minutes of each session) conditions. These conditions were split into blocks of 30 strides, and for each block the outcome measures were computed. For our main analysis, to test whether a training effect occurred across sessions and weeks, for the normal walking and training conditions, we averaged each outcome measure over all 30-stride-blocks within each condition. To assess what changes within a single session underlie these training effects (see S2 File), we used the outcome measures computed from the first 30 strides (i.e. normal walking/training/after-effect start) and the last 30 strides of the condition (i.e. normal walking /training/after-effect end).

## Statistics

**Clinical outcome measures.**   For the clinical outcome measures (SPPB and FES-I scores), we used paired samples t-tests to assess changes between the first session of the control week (baseline) and the end of the last training session. In addition, if a significant effect was found, we assessed whether the change in SPPB/FES-I score was retained at the retention test. To this end, we used a paired samples t-test to test the score at retention against the baseline score from the first session of the control week.

**Biomechanical outcome measures.**   A repeated-measures ANOVA was used to test whether the degree of foot placement control improved during the normal walking condition, as a function of Week ("1","2","3","4") and Session ("1","2"). We chose to test the normal walking condition to represent normal walking rather than the after-effect condition, to avoid confounding by potential immediate after-effects. By adding both "Week" and "Session", we essentially include two factors that both represent time. However, we found that this two-factor model fitted the data better, since there was a relatively short time between "Sessions" within a week, and a bit longer time between the session at the end of the week and the subsequent session in the new week. If "Week", "Session" or their interaction was significant, we conducted Bonferroni corrected post-hoc analyses to determine which comparisons were significant.

To better understand how potential training effects were elicited, we performed another repeated-measures ANOVA. This ANOVA included the factors Week ("2","3","4") and Session ("1","2"), and tested the outcome measures during the training. Here, we focused specifically on any Session effects, since that would demonstrate changes whilst walking on the same LesSchuh (i.e. 2, 1.5 or 1 cm). If "Session" or the "Session*Week" interaction effect was significant, we performed Bonferroni corrected post-hoc analyses. For those interested, in S2 File we

zoomed in even further and tested for immediate and after-effects of walking with LesSchuh, as in Hoogstad, van Leeuwen [21].

The same analyses were applied for the magnitude of foot placement error and the secondary outcome measures. In addition, we tested for retention of those outcome measures that had changed as a function of Week during the normal walking condition, and as a function of Session in the training condition. These retention tests were conducted as paired-samples t-tests either between the first session and retention (normal walking) or between the last training session and retention (LesSchuh walking).

## Results

All participants completed all training sessions and the retention test. Most participants were able to walk on the one-centimeter ridge in the final training week. For one participant, we decided to use the one-and-a-half-centimeter ridge in the final training session and retention test. For this participant, the first session with the one-centimeter ridge proved too challenging to perform within the boundaries of our instructions (i.e. pointing the feet forward, and only letting the ridge touch the floor, not the other parts of the sole). The tables of the statistical tests can be found in S1 File.

### Participant characteristics

Our participant group consisted of fit older adults (SPPB, Table 1), with a low concern of falling (FES-I, Table 1).

### Clinical outcome measures–Training effects

Despite a visible increase in Fig 4, the SPPB score at baseline was not significantly different from the SPPB score at the last training session. Neither was there a significant change in the FES-I score between baseline and the last training session (Fig 5) For more insight into the walking test of the SPPB we have added an overview in S1 Fig.

**Table 1. General assessment\*.**

| Participant characteristics | | |
|---|---|---|
| Age | 73.4 (SD = 5.7) years | |
| Height | 179.7 (SD = 8.9) cm | |
| Weight | 74.5 (SD = 8) kg | |
| Preferred treadmill walking speed | 3.2 (SD = 1.0) km/h | |
| MMSE score | 29.2 (SD = 1.1) | Above inclusion threshold |
| mean SPPB score (baseline) | 11.2 (SD = 1.1) | No risk of impaired physical functioning |
| FES-I score (baseline) | 19 (SD = 2.5) | Low concern of falling |

MMSE: Mini Mental State Examination

SPPB: short physical performance battery

FES-I: Falls Efficacy Scale International

\*An overview of the individual SPPB and FES-I can be found in S1 Fig.

The results of the additional questionnaire can be found as S3 File. Not all participants experienced a subjective training effect, but others were enthusiastic about the training.

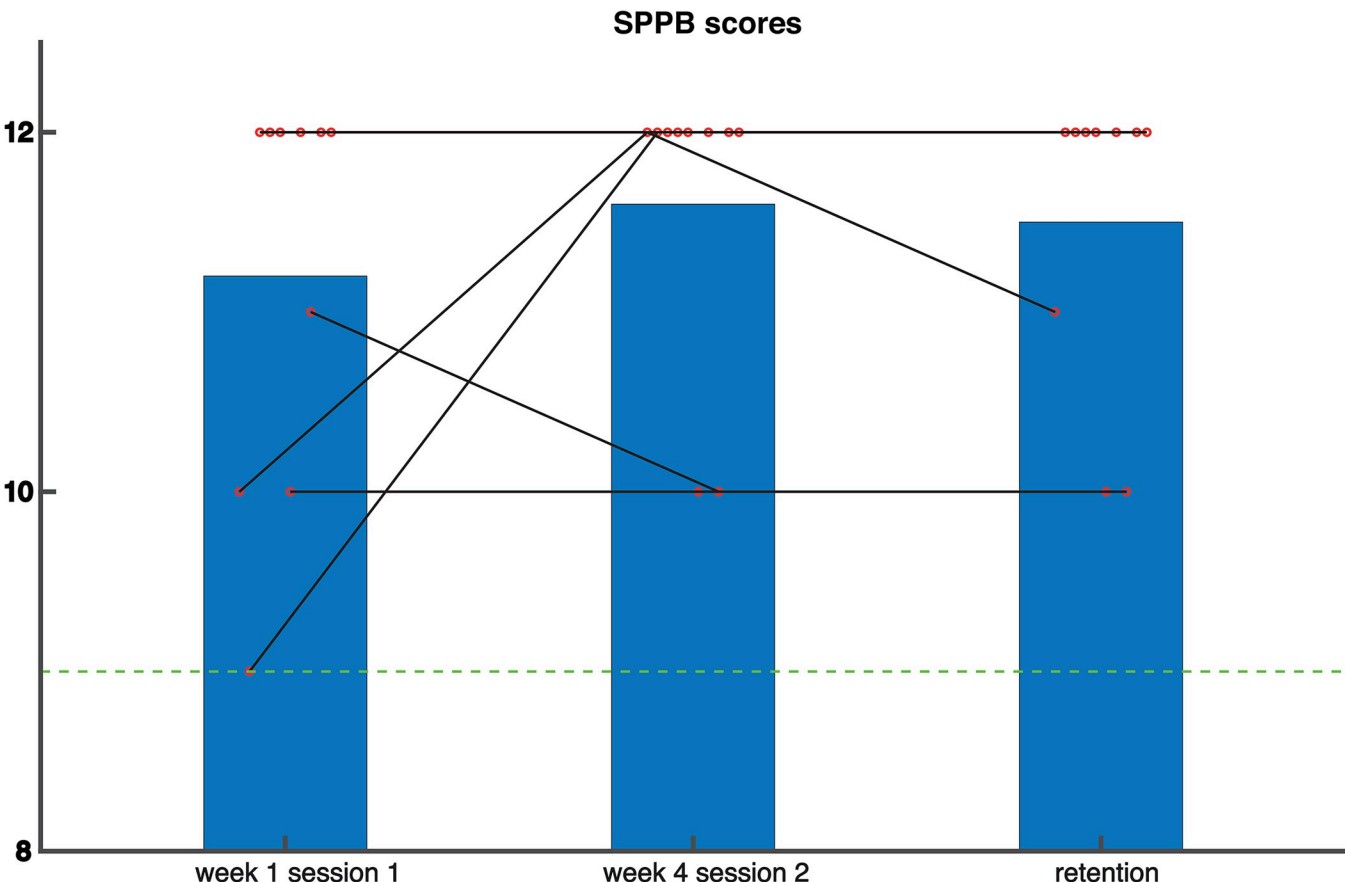

**Fig 4. Scores on the Short Physical Performance Battery (SPPB).** Higher scores represent better physical performance. Red circles represent individual data points. The green line represents the threshold [30] above which participants are in the safe zone.

### Biomechanical outcome measures–Training effects

**Degree of foot placement control—Normal walking.** The degree of foot placement control during the normal walking condition did not significantly change between Sessions or Weeks. Yet, the Week * Session interaction was significant. However, Bonferroni corrected post hoc analysis of the interaction effect did not yield any significant comparisons (Fig 6A).

**Degree of foot placement control—Training condition.** The degree of foot placement control during the training condition, was not significantly affected by either Week or Session, nor by their interaction (Fig 6B).

**Magnitude of foot placement error—Normal walking.** For the normal walking condition, we found a significant effect of Week on the foot placement error. This effect denoted that from week 2 onwards, the foot placement error was smaller than in week 1 (Fig 7A). In weeks 2 and 3 foot placement error had diminished by one millimeter and by two millimeters in week 4, as compared to the control week.

**Magnitude of foot placement error—Training condition.** For the foot placement error during training, we found a significant effect of Session. This signifies that every second time participants trained with the same shoe (respectively 2, 1.5 or 1 cm), the foot placement error was smaller (Fig 7B). The difference between sessions within a week was half-a-millimeter.

**Gait stability—Normal walking.** In the normal walking condition, gait stability was significantly affected by Week and Session (p<0.05). Post-hoc comparisons for Week showed

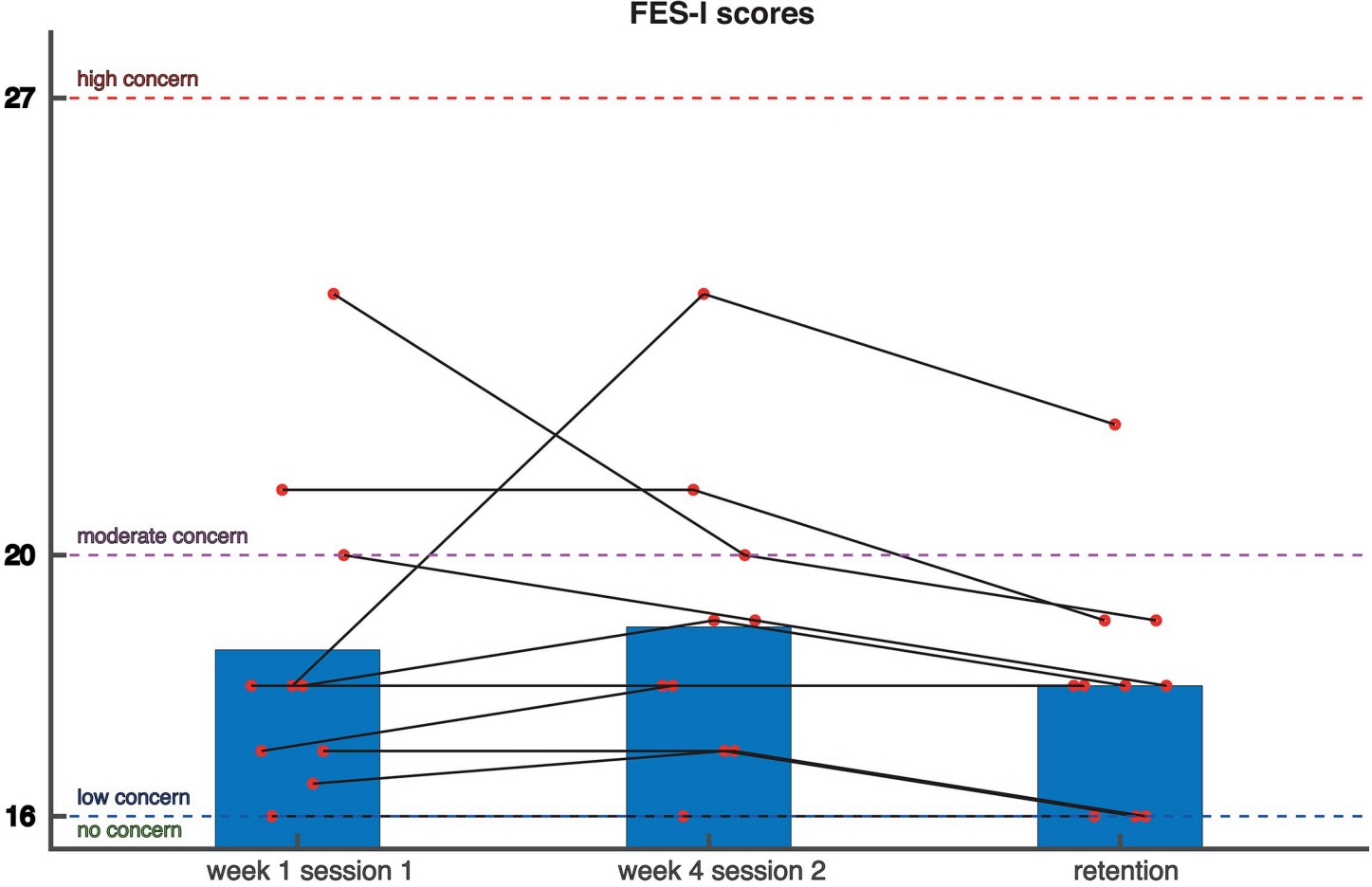

**Fig 5. Scores on the Falls Efficacy Scale-International (FES-I).** Red circles represent individual data points. Higher scores mean a more serious concern of falling.

that from week 3 onwards, participants walked more stable than in week 1. The effect of Session revealed that in every second session, participants walked more stable than in the first session that week (Fig 8A).

**Gait stability—Training condition.** For the training condition, we did not find a significant effect of Week, nor of Session (Fig 8B).

**Step width—Normal walking.** In the normal walking condition, step width was significantly affected by Week. Post-hoc analysis revealed that only week two differed significantly from week one, with a smaller step width in week 2 (Fig 9A).

**Step width—Training condition.** During training, step width did not significantly change across Week and Sessions (Fig 9B).

**Stride time—Normal walking.** During normal walking, there was no significant effect of Week nor Session on stride time (Fig 10A).

**Stride time—Training condition.** For the training condition no significant effects were found for Week nor Session on stride time (Fig 10B).

## Retention

For the retention test we only statistically tested outcome measure that had changed significantly between Weeks (for the normal walking condition) or Sessions (for the training condition).

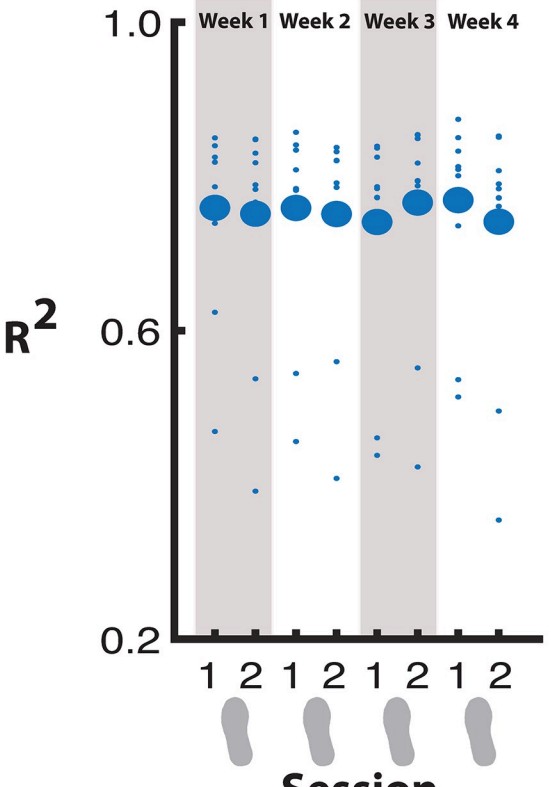
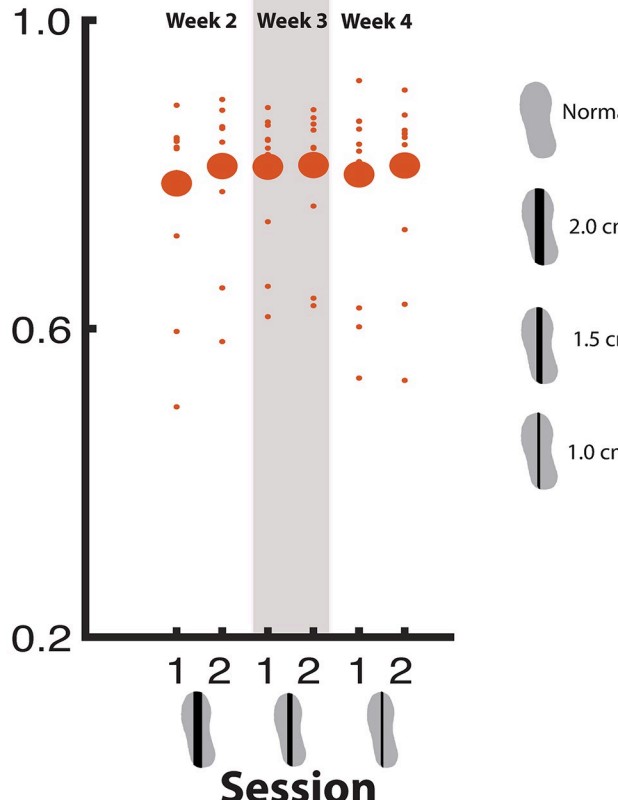

**Fig 6. The degree of foot placement control across measurement sessions.**

**Magnitude of foot placement error.** Foot placement error in the normal walking condition remained significantly smaller during retention as compared to the normal walking condition in week 1. Moreover, when comparing the retention training condition to the training condition of the last training session, foot placement error seemed to have further decreased when walking with LesSchuh (Fig 11), but this was not a significant decrease.

**Gait stability.** During the normal walking condition of the retention session, gait stability was no longer significantly improved compared to the normal walking condition in week 1 (Fig 12).

## Overview main results

In Fig 13 below, we summarize our main findings, i.e. the (absence of) training effects on foot placement control and gait stability, as observed during normal walking and during the training condition.

## Discussion

Maintaining mediolateral gait stability requires accurate coordination between the center of mass (CoM) and foot placement. Here we tried to train foot placement control in older adults, to improve their mediolateral gait stability. We expected that walking with constrained ankle moments would enforce participants to increase their degree of foot placement control, as they

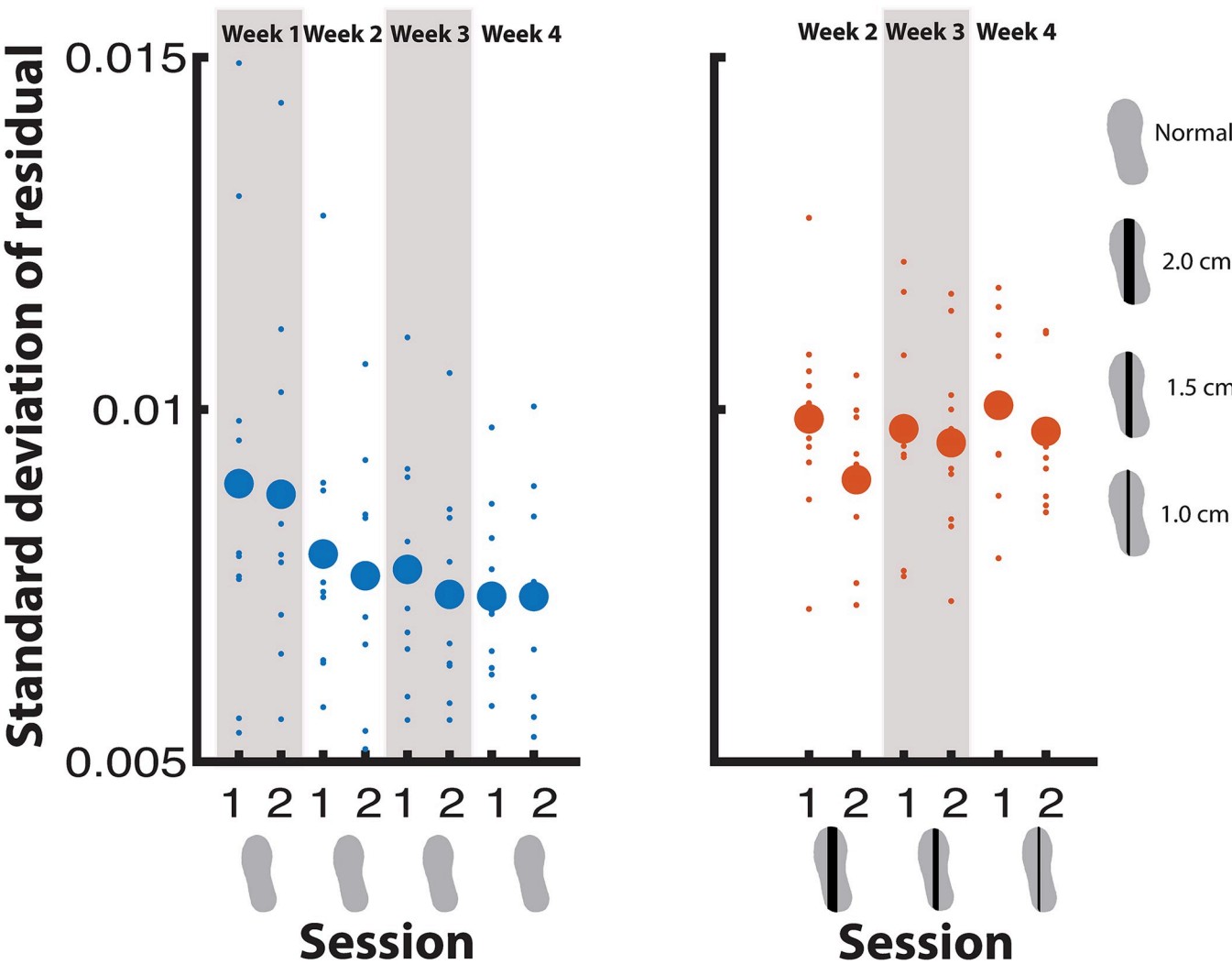

**Fig 7. Magnitude of foot placement error (i.e. standard deviation of residual) in meters across measurement sessions.**

could no longer rely on ankle moment control to compensate for errors in foot placement. Although we did not find the hypothesized changes in the degree of foot placement control, foot placement errors decreased. Moreover, gait stability improved across training sessions and weeks, albeit without retention. Below we will discuss possible explanations as to why the ankle moment constraints did not induce changes in the degree of foot placement control, and how other mechanisms may have contributed to their improved gait stability.

### Foot placement control

Before we dive further into the discussion of our results, we first want to emphasize that we interpret the foot placement control model (Model 1) as a feedback mechanism in which sensory information CoM kinematic state [12], is used to predict a target foot placement location (FP), that will ensure mediolateral gait stability [6]. During swing, the swing leg is steered

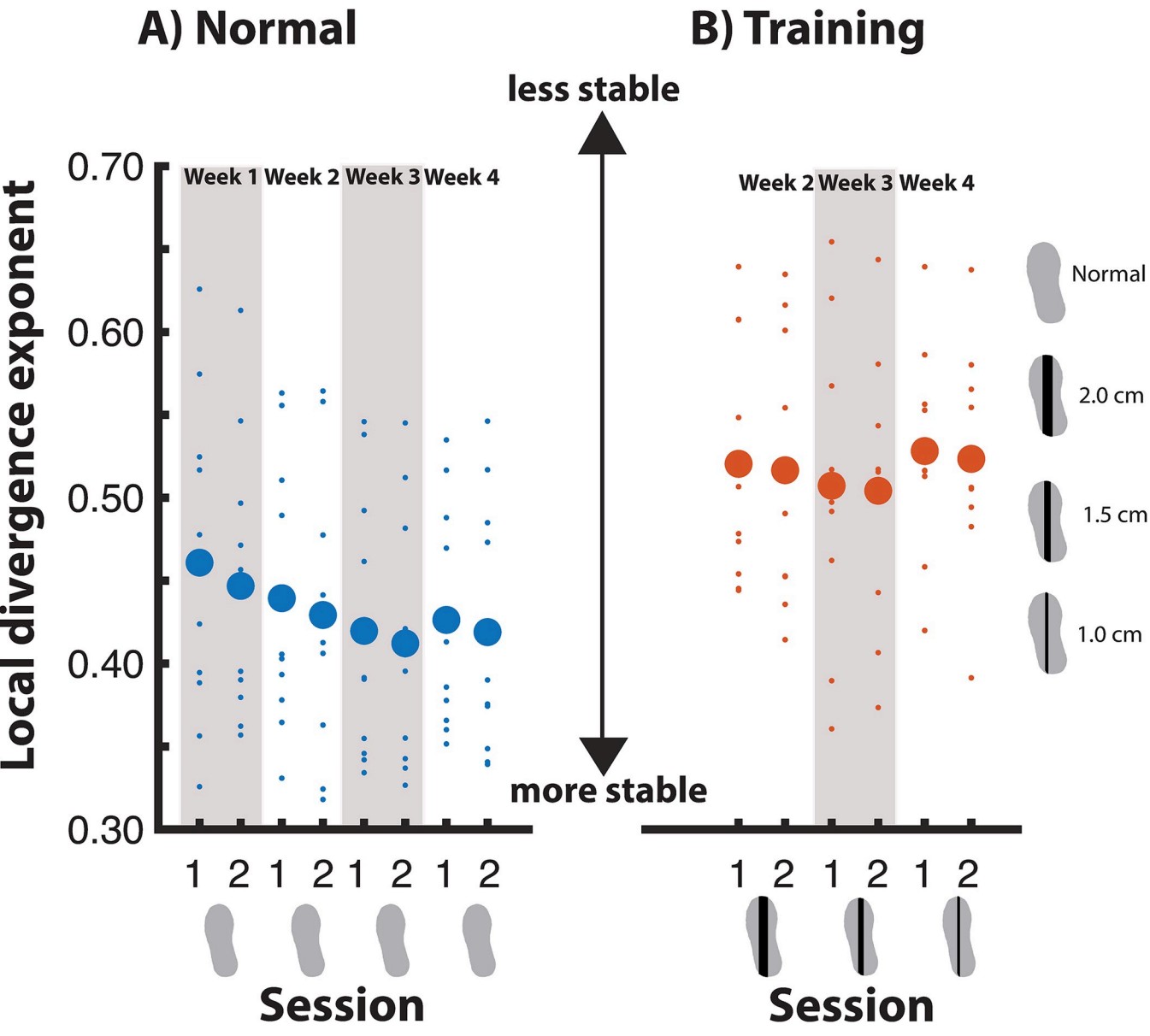

**Fig 8. Gait stability across measurement sessions.**

towards this target location [13, 18], however, once the foot is placed, actual foot placement does not exactly match the predicted/target foot placement location (FP) [7].

The residuals of Model 1 denote these foot placement errors and can be considered as the result of sensorimotor noise or loose control. Decreasing these foot placement errors would mean more precise foot placement control, and this could be an aim for training.

The degree of foot placement control can be quantified as the relative explained variance ($R^2$) of Model 1. A reduction in foot placement error could result in a higher $R^2$, i.e. a higher degree of foot placement control. However, as illustrated in Fig 14, apart from changes in foot placement error, other factors can influence the $R^2$ as well. The changes in Fig 14 panels C-D result in an increase in absolute explained variance. In earlier work, we referred to a higher absolute explained foot placement variance as an increase in the "foot placement contribution" [21].

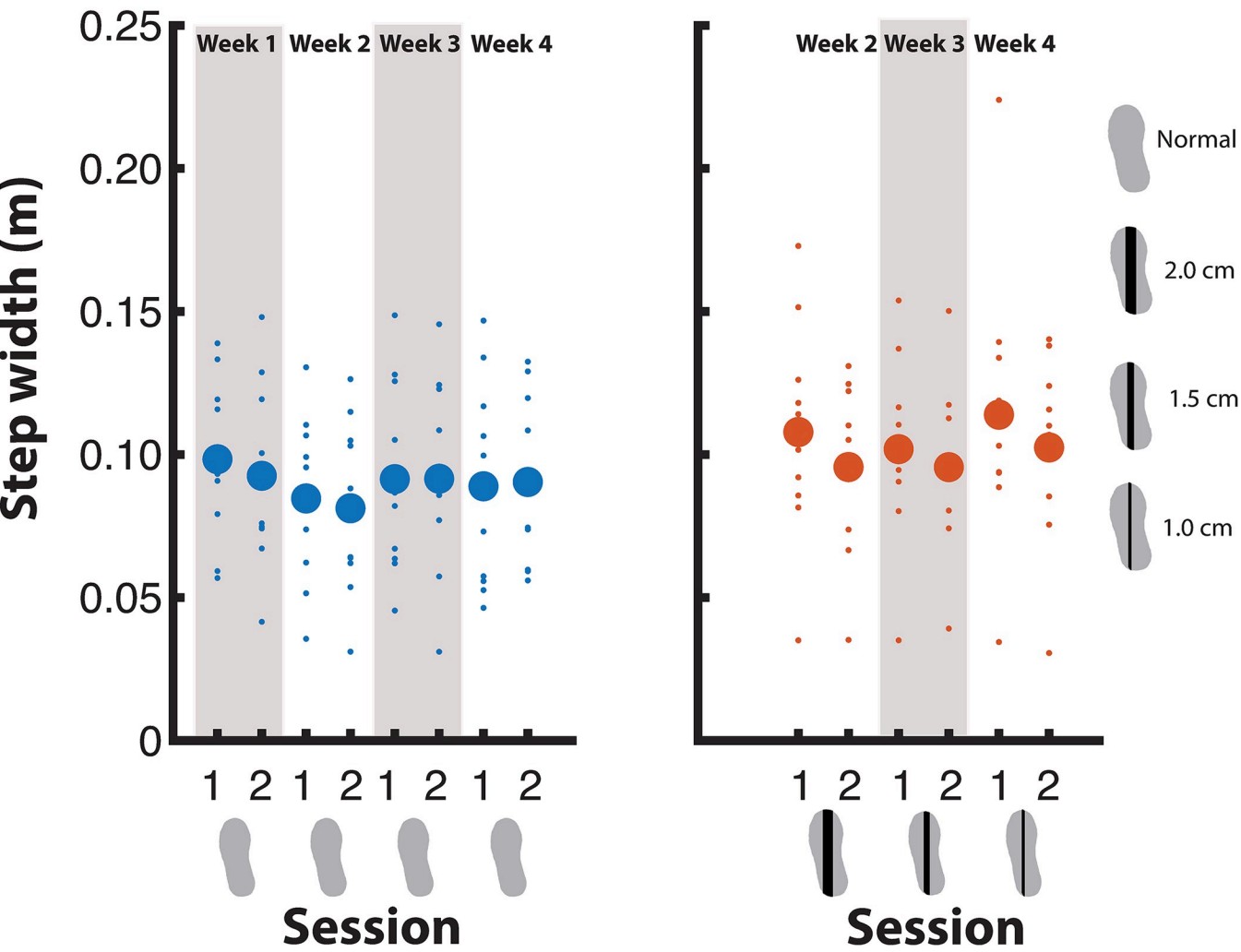

**Fig 9. Step width across measurement sessions.**

A higher foot placement contribution could be an aim for training, if one would like to alleviate other stability control mechanisms. For example, increasing foot placement contribution allows for more CoM kinematic state variability during swing, and perhaps less use of an energetically costly ankle mechanism [31]. And, for instance, an increased step-by-step foot placement contribution may also allow for less wider steps [32], again potentially decreasing energy cost [33]. On the other hand, one could argue that stability training should be aimed at reducing CoM kinematic state variance, and this may be reflected by a lower absolute foot placement contribution instead (see Fig 14 panel C).

The degree of foot placement control ($R^2$) takes both the absolute explained variance and the foot placement error into account, giving a good summary of foot placement control as a stability mechanism. However, only with an absolute measure of foot placement error can

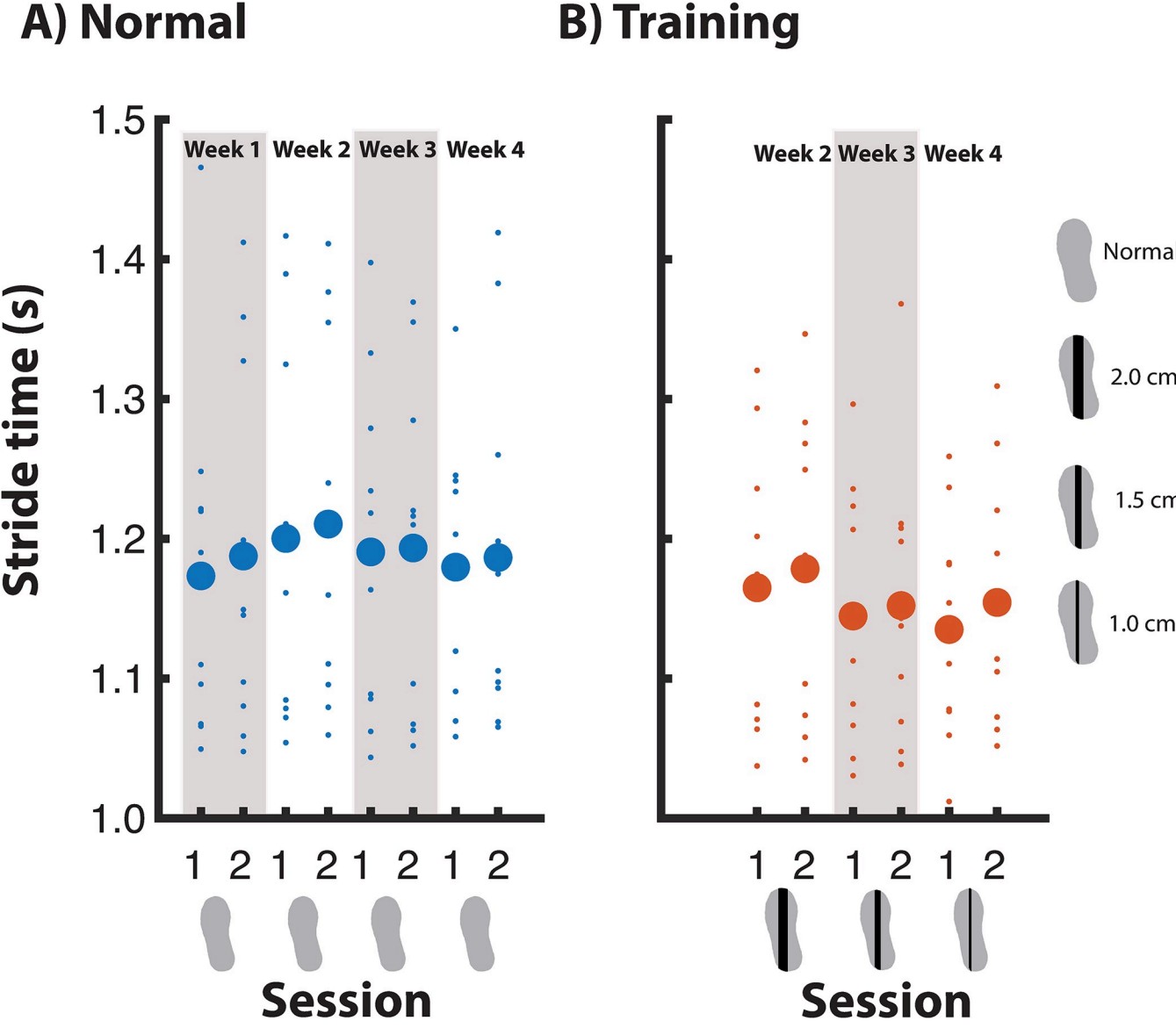

**Fig 10. Stride time across measurement sessions.**

inferences be made regarding (improved) foot placement precision. A high absolute explained variance may lead to an overestimation of foot placement precision, when only considering the $R^2$. In the other way around, a low absolute explained variance may lead to an underestimation of foot placement precision. Moreover, if the absolute explained variance and the foot placement error both increase/decrease, these changes may be hidden behind a constant $R^2$, as is the case in the current study.

## Task compliance

LesSchuh's main function is to limit center of pressure (CoP) adjustments, after the foot is placed. As such, errors in foot placement can no longer be corrected for by ankle moment control, as occurs in unconstrained steady-state walking [20]. Unlike the young adults from our

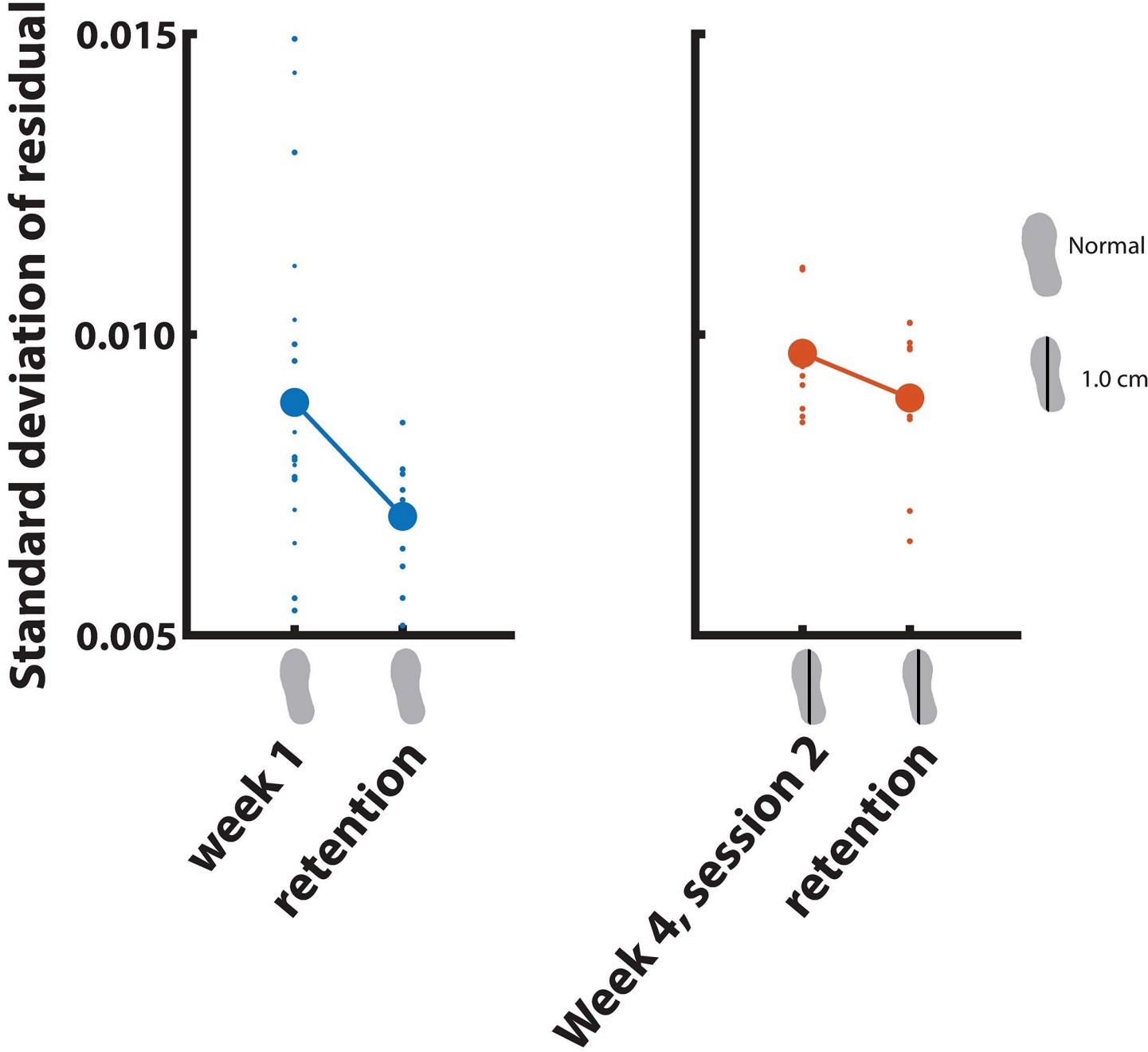

**Fig 11. Magnitude of foot pacement placement error in meters retention.**

previous study [21], as a group, the older adults complied with our instructions to keep their feet pointing straight ahead (see S2 File). As such, they avoided compensation through a toe-ing-out strategy [26]. Thus, despite failing to walk only on the ridge from time to time, corrected by our feedback, the participants managed to walk in such a way that overall ankle

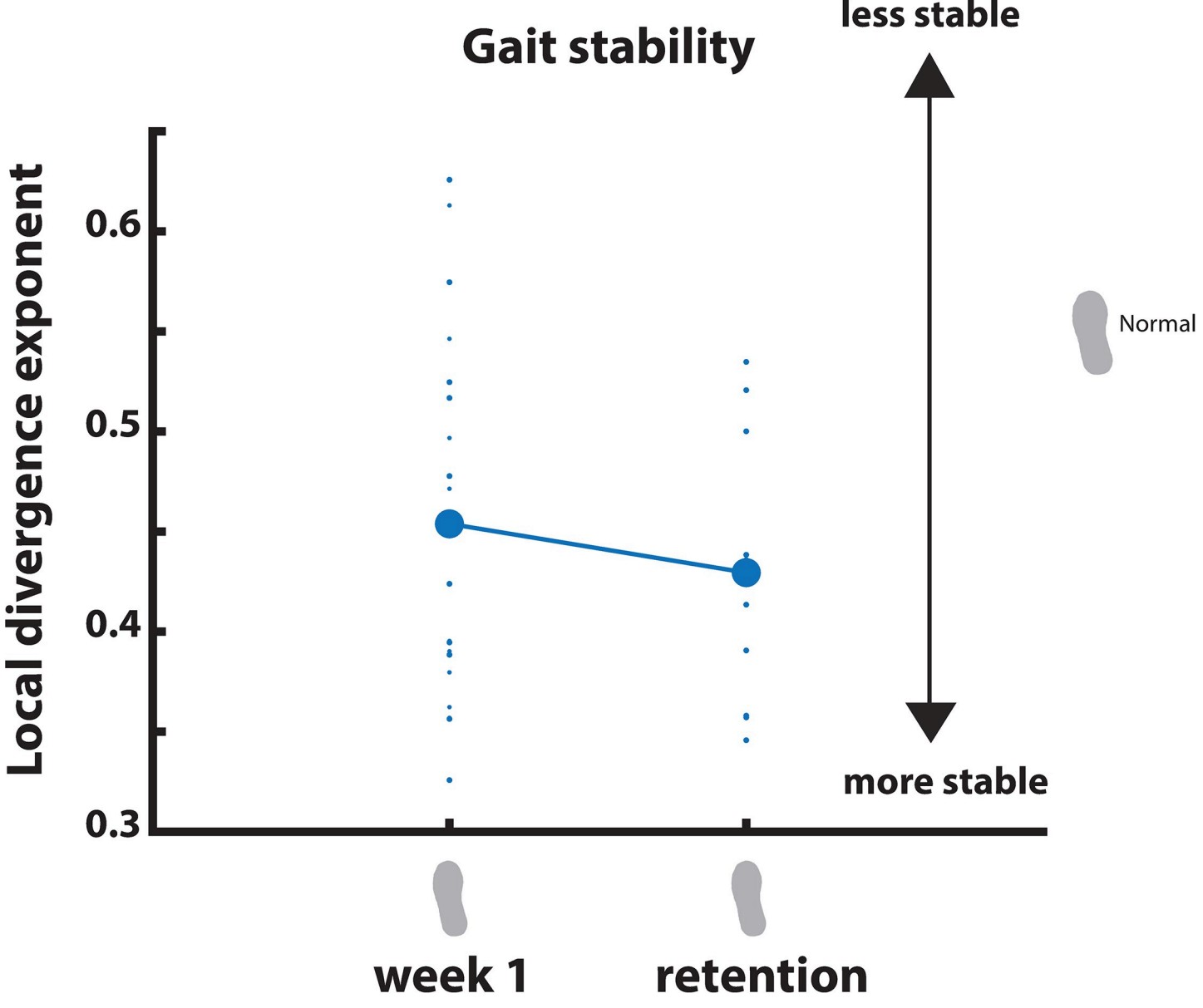

**Fig 12. Gait stability retention.**

moment control was constrained. Non-compliance with the instructions is therefore not a likely explanation for the absent change in the degree of foot placement control. It should be noted however, that even when participants fully complied with the instructions, we did not control the use of other stabilizing mechanisms, such as an angular momentum mechanism [9, 34], or modulation of stride frequency [35].

### Training effects

Although we did not find a training effect for the degree of foot placement control, the older adults demonstrated other training effects across sessions and weeks. Most of these effects were found during normal walking, rather than while walking on LesSchuh.

| | | Normal walking | | | Training condition | | |
|---|---|---|---|---|---|---|---|
| | | Session effect | Week effect | Retention | Session effect | Week effect | Retention |
| Foot placement control | Degree of foot placement control ($R^2$ Model 1) | - | - | *Not tested* | - | - | *Not tested* |
| | Foot placement error (residual model 1) | - | ✓ | ✓ | ✓ | - | *Non-significant trend* |
| Stability | Local divergence exponent | ✓ | ✓ | - | - | - | *Not tested* |

**Fig 13. Main results: Training effects on foot placement control and gait stability.**

Foot placement error during the normal walking condition was decreased relative to week 1 from week 2 onwards, but did not show any further significant reduction over the weeks that followed. From Fig 7, it seems that part of the reduction can be attributed to treadmill walking in itself (large drop between week 1 session 2 and week 2 session 1), possibly reflecting familiarization [36]. However, LesSchuh may have played a role in this training effect as well. In Fig 7, we can see that for the normal walking condition, the largest decrease in foot placement error was found between the last session of the control week and the start of the first training week. However, within the first training week (week 2), a further reduction in foot placement error is observable. As we found significant reductions in foot placement error across sessions during the training condition, one may speculate that these changes during the training were, to some extent, transferred to normal walking. That we do not find a decrease in foot placement error during training across weeks, may be explained by the fact that the ridge underneath the shoe became narrower in each training week. This increased task difficulty, which would explain why in the first session of each training week the foot placement error was increased, while it was decreased again in the second session (Fig 7).

For step width and stride frequency (Figs 9 and 10), we also noticed an increase in the training condition of the week's first session, followed by a decrease in the second session. Although

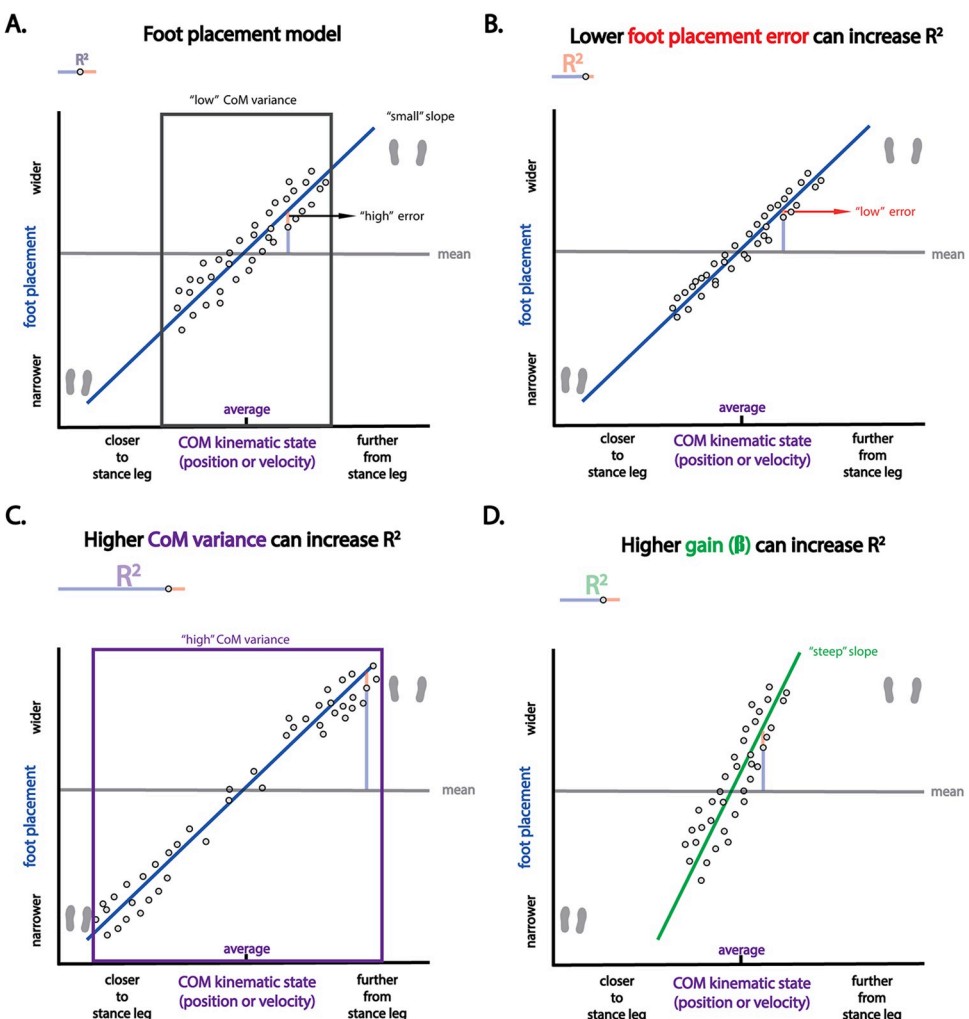

**Fig 14. The degree of foot placement control.** The degree of foot placement control ($R^2$) can be altered in different ways. In all panels we see fictive data of the foot placement model. As a simplification we have drawn the linear relationship for one of the predictors (it can be either CoM position or velocity). Panel A shows a foot placement model with a "high" foot placement error, a "low" CoM variance and a "small" slope. In panels B-D, we illustrate respectively how a "low" error, a "high" CoM variance and a "steep" slope increase the $R^2$ of this model.

these trends were not significant, this could reflect compensation for the ankle moment constraints, and a lesser need for compensation in the second training session with the same shoe, as foot placement error decreased. In previous studies, increased step width and decreased stride time were found to be used to compensate for the constrained ankle moments [18, 21]. However, in the current study such compensations (see "Immediate effect of walking with ankle moment constraints", S2 File) were not significant, nor was there a significant effect of Session on step width and stride time. The use of a single-belt treadmill in the current study, rather than a split-belt, may have diminished the need for such compensatory strategies [19].

Another training-effect was found for gait stability. We defined gait stability as a local divergence exponent, which has previously been shown to be able to distinguish fall-prone from healthy older adults [37]. During the normal walking condition, gait stability consistently improved between sessions, and from week 3 onwards participants walked more stable as compared to the control week. It is likely that part of the improvement in gait stability can be

attributed to the LesSchuh training, rather than just familiarization, since the first significant difference between weeks occurred after LesSchuh training. An alternative explanation that cannot be ruled out however, would be that treadmill familiarization occurs gradually across weeks, which for the local divergence exponent becomes apparent from week 3 onwards. Although several gait parameters have shown to stabilize after only six minutes of treadmill walking [36], it is still unclear after how much normal treadmill walking the local divergence exponent would demonstrate plateau values.

## Stabilizing strategy

From our outcome measures, the reduction in foot placement error is the most likely candidate to underlie the enhanced gait stability. Since none of our outcome measures exactly mirrored the changes in stability, it is likely that other strategies outside the scope of our outcome measures, such as an angular momenum strategy [9, 34], contributed to the improved gait stability.

Intuitively, one would expect that a decrease in foot placement error would coincide with a higher degree of foot placement control. However, going back to Fig 14, we can see that this is not necesarily the case. Whereas a lower foot placement error can increase the degree of foot placement control, a lower CoM variance can decrease it, by lowering the absolute explained foot placement variance (i.e. the foot placement contribution). Given the unchanged degree of foot placement control, in combination with the reduction in foot placement error, we can infer that the foot placement contribution decreased. Possibly, this reduction in foot placement contribution is caused by the more precise foot placement. More precise foot placement in a given step would better attenuate the deviations in CoM kinematic state and thus need less foot placement adjustment on the next step. As such, a lower absolute (explained) foot placement variance could result from reducing foot placement errors. Alternatively, other stability control mechanisms may have contributed, allowing for lower foot placement variance.

It should be noted that for our participants it was quite challenging to support themselves on the narrow ridge underneath the shoe. This was not necessarily only related to the limited CoP shift, but, based on their feedback, also on the muscle activity and strength required to keep their feet in the required orientation. This additional challenge may have enhanced their general coordination and muscle strength, although we did not perform measurements to verify this. If so, the training may have facilitated stability control in general, as opposed to a specific adaptation to the lack of a compensatory mechanism for errors in foot placement.

## Retention

During normal walking in the retention test, the foot placement error remained smaller compared to the control week. Despite a retained reduction in the magnitude of foot placement error, stability had returned to the control week level in the retention test. It seems that participants needed to keep training/walking on the treadmill to retain improved stability. The mismatch in the retention results of foot placement error and gait stability is unexpected. Improved foot placement error was expected to lead to improved gait stability. However, as mentioned above, none of our outcome measures exactly mirrored the changes in stability, suggesting that other stability control mechanisms play a role as well. We speculate that, although foot placement was tightly controlled (as reflected by the decreased foot placement error) during the retention test, other mechanisms could have been less used.

## Training potential

Although this study included a fit group of older adults (based on the SPBB), the training proved quite strenuous for some participants. Compared to young adults, the older adults

needed more intensive feedback when walking on LesSchuh. Since for this fit and physically active group it was already a challenge to stay on the ridge, often resulting in muscle soreness on subsequent days, the applicability may be limited. Yet, overall our participants expressed positive feelings regarding the training, and one participant even took LesSchuh home to continue the training. If LesSchuh can effectively enhance stability, the fact that they can easily be made using low-cost materials and the ability to train at home promote the potential as a training tool. As such, LesSchuh could be suitable as a training tool for relatively fit older adults. However, for frail older adults more controlled training methods, such as assistive force fields [38, 39] or augmented proprioceptive feedback [40], may be preferred in terms of effort and feasibility.

## Study limitations

We only included a small sample size with relatively fit older adults. At baseline, on average the group had "No risk of impaired physical functioning" (SPPB) and had a "Low concern of falling" (FES-I). Perhaps the high scores on both tests at baseline, prevented a significant improvement on these tests. A larger, more diverse group, in terms of SPPB and FES-I levels, may have given more insight into a potential effect of LesSchuh on these clinical measures. Another limitation of this study is, that it is not a randomized control trial, and therefore it cannot be concluded whether the training effects should be attributed to training on LesSchuh, or to treadmill walking in itself. We did have the control week, theoretically as a control for within week (Session) effects. During normal walking, stability did not only improve across sessions in the training weeks, but also in the control week, likely reflecting treadmill familiarization [36]. This makes it hard to distinguish true LesSchuh training effects on stability. The presence of a control group would have alleviated confounding effects of familiarization, and its absence is a clear limitation of this study. In future work, a randomized controlled trial, with normal treadmill walking as a control intervention could verify the presumed effects of LesSchuh. Moreover, although this study showed training effects in normal treadmill walking, it remains to be elucidated whether such training effects translate to overground walking.

Lastly, future studies are recommended to collect full body kinematics, center-of-pressure data and energy cost data to be able to investigate whether apart from desired changes in foot placement and stability control, no maladaptive changes are found in the gait pattern, following training with LesSchuh. A potential maladaptive change can be related to potential stiffening of the ankle joint in order to support oneself only on the narrow ridge. Increasing ankle stiffness is a part of ageing [41], and may not only be detrimental to gait stability, but also to an energetically inefficient push-off mechanism [42]. In future studies, it should be monitored whether LesSchuh teaches a gait pattern in which even less ankle range of motion is used, and whether this pattern is transferred to normal walking, as such undesired effects should be avoided. From an optimistic perspective, the stiffening of the ankle and the effort exerted to stay on the ridge may instead have a positive effect on ankle functionality, as muscle strength exercises are recommended to combat ankle stiffness, in combination with stretching exercises (Vandervoort, 1999). Perhaps combining LesSchuh training with stretching exercises will prevent maladaptive changes related to ankle stiffness.

## Functional relevance

The largest training effect was observed during normal walking in week 4. In this week, foot placement error had reduced two millimeters compared to the control week. Foot placement is an effective stability mechanism as it largely determines the center-of-pressure position, and consequently the resulting moment of the ground reaction force, which accelerates the

CoM in the mediolateral direction [9, 43]. To estimate of a two millimeter foot placement error on the moment generated, we consider the vertical ground reaction force, and the center-of-mass as the pivot. The mediolateral distance between the projection of the center-of-mass on the floor and the center-of-presssure then defines the moment arm. We thus have:

$$M = F_{grfvert} * (CoP_{MLaxis} - CoM_{MLaxis)}$$

In which $M$ is the moment generated by the vertical ground reaction force, $F_{grfvert}$ is the vertical groun reaction force, $CoP_{MLaxis}$ is the mediolateral center-of-pressure position, and $CoM_{MLaxis}$ the mediolateral CoM position, both with respect to the other stance foot.

Rounded up our participants had an average weight of 75 kilogram. So we will assume a vertical ground reaction force of 750 Newton. The participants walked with step widths around 0.10 meter. For the sake of simplicity, we assume that the center-of-mass is positioned in between the two feet. If this is the case, the average moment arm for our moment calculation is:

$$CoP_{MLaxis} - CoP_{MLaxis} = 0.1 - 0.05 = 0.05 \, m$$

And the average moment would be:

$$M = F_{grfvert} * (CoP_{MLaxis} - CoM_{MLaxis}) = 750 * 0.05 = 37.5 \, Nm$$

A foot placement error of two millimeters too medial, and the same $F_{grfvert}$, would computed:

$$M_{error} = F_{grfvert} * (CoP_{MLaxis} - CoM_{MLaxis}) = 750 * 0.048 = 36 \, Nm$$

Thus, a corrective moment of 1.5 Nm would be needed to compensate for a foot placement error of two millimeters.

Although the difference between the two moment calculations ($M$ and $M_{error}$) does not sseem very large, after erronous foot placement, a fast corrective response would call uupon an ankle or angular momentuum mechanism (i.e. the other two analytically distinguishable components contributing to the mediolateral CoM acceleration [9, 43]). During gait, stabilizing moments can be genereated relatively energetically inexpensively through foot placmeent control (only the weight of the swing leg needs to be moved to prevent foot placement errors) [31]. Therefore, even a two-millimeter reduction in foot placement error may be beneficial. Apart from energy efficiency, during steady-state walking, we can be at risk of falling sidewards if our center-of-mass moves close to the lateral border of our base of support. When near this border, even small corrections that help the CoM to stay just medially of this border may prevent a side-ward fall.

## Conclusion

Older adults have a lesser degree of foot placement control than young adults [12], which can be detrimental to mediolateral gait stability [6]. Therefore, we set out to train their degree of foot placement control by constraining ankle moments. Their degree of foot placement control remained unchanged, yet foot placement errors decreased and gait stability increased. We conclude that decreased foot placement errors and increased gait stability during normal walking may be explained by familiarization to treadmill walking and training with the LesSchuh. The extent to which familiarization and LesSchuh training contribute requires a control study.

## Supporting information

**S1 Fig. SPPB four meter walk test.**
(PDF)

**S1 File. Table main analysis statistical tests.**
(PDF)

**S2 File. Within session analyses.**
(PDF)

**S3 File. Quetionnaires.**
(PDF)

**S4 File. Inclusion/exclusion questionnaire.**
(PDF)

## Acknowledgments

The researchers would like to express their gratitude to all participants. Not only are we thrilled that all of them completed the training schedule, but it was also a pleasure to meet you all and listen to your stories. Also thank you Mirjam Pijnappels for providing the inclusion/exclusion questionnaire.

## Author Contributions

**Conceptualization:** Mohammadreza Mahaki, Anina Moira van Leeuwen, Sjoerd M. Bruijn, Jaap H. van Dieën.

**Formal analysis:** Anina Moira van Leeuwen.

**Investigation:** Mohammadreza Mahaki, Anina Moira van Leeuwen.

**Methodology:** Mohammadreza Mahaki, Anina Moira van Leeuwen, Nathalie van der Velde, Jaap H. van Dieën.

**Supervision:** Sjoerd M. Bruijn, Nathalie van der Velde, Jaap H. van Dieën.

**Visualization:** Mohammadreza Mahaki.

**Writing – original draft:** Anina Moira van Leeuwen.

**Writing – review & editing:** Mohammadreza Mahaki, Anina Moira van Leeuwen, Sjoerd M. Bruijn, Nathalie van der Velde, Jaap H. van Dieën.

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
