## [Decision Letter · Decision Letter 0]

25 May 2023

PONE-D-23-09453Foot placement control can be trained: Older adults learn to walk more stable, when ankle moments are constrainedPLOS ONE

Dear Dr. van Leeuwen,

Thank you for submitting your manuscript to PLOS ONE. After careful consideration, we feel that it has merit but does not fully meet PLOS ONE’s publication criteria as it currently stands. Therefore, we invite you to submit a revised version of the manuscript that addresses the points raised during the review process.

Thank you for submitting your manuscript to PLOS ONE. Two reviewers viewed this work favorably and have provided comments and suggestions on the manuscript. Please be sure to provide point-by-point responses to all reviewer comments along with your revised manuscript.

We look forward to receiving your revised manuscript.

Kind regards,

Ryan Thomas Roemmich

Academic Editor

PLOS ONE

Additional Editor Comments:

Thank you for submitting your manuscript to PLOS ONE. Two reviewers viewed this work favorably and have provided comments and suggestions on the manuscript. Please be sure to provide point-by-point responses to all reviewer comments along with your revised manuscript.

Reviewers' comments:

Reviewer's Responses to Questions

**Comments to the Author**

1. Is the manuscript technically sound, and do the data support the conclusions?

Reviewer #1: Yes

Reviewer #2: Yes

2. Has the statistical analysis been performed appropriately and rigorously? 

Reviewer #1: Yes

Reviewer #2: Yes

3. Have the authors made all data underlying the findings in their manuscript fully available?

Reviewer #1: Yes

Reviewer #2: Yes

4. Is the manuscript presented in an intelligible fashion and written in standard English?

Reviewer #1: Yes

Reviewer #2: Yes

5. Review Comments to the Author

Reviewer #1: The study determines if foot placement control – a major gait strategy – can be trained in older adults. The training constrains the medio-lateral ML ankle moment, such that any corrections are accomplished by foot placement, rather than ML ankle moment adjustments after the foot is in stance phase. ML ankle moment is constrained by special shoes with a very narrow base under the shoe. Training occurred over three weeks, with a baseline/control week preceding training, and a post-training retention.

I find this approach to training foot placement is innovative and creative. The manuscript is clear, and the study is relevant to both basic scientists and clinicians. The statistics are appropriate. Especially, the figures are very clear with shaded areas, etc. to delineate different aspects of the timeline.

I have a few comments that I would like to see addressed.

1. Please clarify the conceptual differences between foot placement control and foot placement error. It appears logical to expect that if foot placement control is impaired, then foot placement error will increase, and vice versa. In the current results, there was no change in foot placement control, but foot placement error decreased. While the explanation on lines 456+ did help somewhat, I was left feeling dissatisfied with the explanation. The explanation was based on the model, but a physiological rationale was not provided. The explanation indicates that the lower magnitude of foot placement control may result from less error (i.e., less variance) in foot placement (lines 460-462). This argument seems problematic if it implies that large errors will result in large amount of control. Further clarification would be helpful. Similarly, the observation that stability changes did not persist at retention, but foot placement error did persist appears inconsistent and also left me feeling dissatisfied.

2. Please address the role of ML moments of the body besides the ankles. While the ankle ML moments were constrained, other ML moments such as trunk lateral flexors and the hips were not constrained. It seems they could take on (some of) the role of the ankle ML moments.

3. While the result was statistically significant, please address the functional consequence of a 0.5 mm change in foot placement error. For example, can the authors estimate the corrective moment/torque needed to correct for a 0.5 mm foot placement error?

4. Consider adding medio-lateral to the title and abstract for clarity (i.e., ML ankle moments)

5. Some of the observed changes may result from requiring the participants to walk at the their preferred treadmill speed (determined with flat shoes) for all conditions. I expect their preferred gait speed on the training shoes would be slower if they had the option.

6. The total duration of training was 90 minutes, which seems short relative to other training studies. Please comment, especially related to interpretations about strength enhancement (lines 469-470).

7. Line 102 – correct the spelling of ‘anonymously’

8. Text on lines 429+ dicusses the differences in step width and stride time, but they were not significant. This text should be removed or it should be explicitly stated that these are trends when they are first discussed. It’s pointed out later, but currently the text reads as if the values are significantly different.

Reviewer #2: This paper makes a valuable contribution to research that tries to unravel (impaired) mediolateral balance control during walking. Although prior research from this group uses similar paradigms I want to emphasize the their study as a very nice and original example of experimental design. The study would have benefited from a control group, mainly to account for the effects of familiarization to treadmill walking. Nevertheless, this should not hold this work from being published in its current form (set apart some comments the authors can find below), as there are many insights to be gained form the manuscript as it is. The manuscript is very well structured.

1) Explain foot placement error and control by providing some examples.

An important comment that I would like to see addressed is on the explanation of foot placement error and foot placement control. These are the two most important concepts of the study and are not easy to comprehend. I like them as model of quantifying foot placement control. However, it took some time to understand exactly what these mean. I describe three scenarios below to check my understanding with the authors.

If these, or a variation of these, make sense, these could be added somewhere for clarification of this model? The wording can be improved and shortened probably.

• If FP varies a lot but COM motion is always the same, foot placement control is small and the foot placement error is large. Subjects are changing foot placement for no reason?

• If FP variance is large and can be explained by COM motion for 90%, subjects are adequately modulating foot placement to the needs of maintaining stability as COM is variable. Foot placement error is still large as 10% of the large original variance is high and might give balance problems or require an additional mechanism to control for balance.

• If FP variance is small and can be explained by COM motion for 90%, subjects are adequately adapting foot placement to the needs of maintaining stability as COM is variable. Foot placement error is small as 10% of original variance is low and should not give balance problems.

I understand that these concepts might be explained in prior work of the authors. But I think the paper should stand on its own.

2) Motivate the use of foot placement error instead of foot placement variability

When reading the results, I sometimes doubt whether it would be better to just look at foot placement variance instead of foot placement error. Can the authors defend their choice of choosing the placement error as they define it?

3) Can you bring the results together in one overview table/figure.

I think the existing figures can stay as they are. But, since so many results are to be interpreted it would be nice to have an overview table that summarizes what is changing and not changing across week/session. The reader can during reading the discussion than easily refer to this table.

Some other minor comments

Figures: They are very low resolution in the manuscript we received. I assume this will be fixed.

L 102: Anonumously -- > anonymously

L442-446: Or gait stability as calculated here is a more subtle measure that can change for longer due to familiarization compared to foot placement control.

L528: Although the discussion explains all details well it seems that the conclusion might overemphasize the foot placement errors and gait stability. Maybe change to: … decreased foot placement errors and increased gait stability during normal walking may be explained by familiarization to treadmill walking and training with the LesSchuh. The extent to which they contribute requires a control study.

Limitation: Can you mention the lack of a control group as a limitation that could have alleviated the confounding effects of familiarization?

Future work: The authors mention collecting full kinematics for these studies. This would be great for sure (control for and observe inertial compensation strategies). It is worth mentioning that this can be enabled by markerless mocap as I assume the choice to not collect full kinematics in this study was made because of time constraints.

Just a question out of interest: Is there a high risk of spraining your ankle with the LesSchuh?

6. PLOS authors have the option to publish the peer review history of their article (what does this mean?). If published, this will include your full peer review and any attached files.

Reviewer #1: No

Reviewer #2: No

---

## [Author Response · Author response to Decision Letter 0]

7 Sep 2023

Dear reviewers,

We would like to thank you for both expressing enthusiasm as well as sharing critical comments related to our study. We have improved our manuscript based on your comments and have highlighted our changes in the manuscript. Below you will find a point-by-point response to your comments.

5. Review Comments to the Author

Reviewer #1: The study determines if foot placement control – a major gait strategy – can be trained in older adults. The training constrains the medio-lateral ML ankle moment, such that any corrections are accomplished by foot placement, rather than ML ankle moment adjustments after the foot is in stance phase. ML ankle moment is constrained by special shoes with a very narrow base under the shoe. Training occurred over three weeks, with a baseline/control week preceding training, and a post-training retention.

I find this approach to training foot placement is innovative and creative. The manuscript is clear, and the study is relevant to both basic scientists and clinicians. The statistics are appropriate. Especially, the figures are very clear with shaded areas, etc. to delineate different aspects of the timeline.

I have a few comments that I would like to see addressed.

1. Please clarify the conceptual differences between foot placement control and foot placement error. It appears logical to expect that if foot placement control is impaired, then foot placement error will increase, and vice versa. In the current results, there was no change in foot placement control, but foot placement error decreased. While the explanation on lines 456+ did help somewhat, I was left feeling dissatisfied with the explanation. 

What you outline here is the problem of choosing the proper outcome measure. Although Model 1 seems to be a good description of foot placement control based on center-of-mass kinematic state information, it is less clear which outcome measure best describes the quality of this foot placement control. In our earlier work, in line with [1], we focused mostly on the relative explained variance (R2) of this model, which describes the percentage of foot placement variance that is explained by the model, and we referred to it as “the degree of foot placement control”. However, the R2 can change in different ways. It is dependent on the foot placement error (higher error, lower R2), center-of-mass kinematic state variance (higher variance, higher R2) and the regression coefficients of Model 1 (higher gain, higher R2). To illustrate the latter statements, we have now added Figure 13 to the manuscript. 

When considering the R2 alone, we cannot distinguish how these three factors interacted to result in changes in the R2. Or in the case of the current study, based on the R2 alone, we did not find any changes in foot placement control, whereas changes in foot placement error showed there were. Although in earlier work we have provided similar information in the supplementary material [2], this is the first study where we have added foot placement error as an outcome measure. We have now learned that a combination of outcome measures (in this case both R2 and foot placement error) is needed to gain insight into how foot placement control changes across conditions. Here we chose to add foot placement error next to the R2 measure to, apart from a relative degree of foot placement control, also consider whether foot placement error decreased in absolute sense.

Apart from Figure 13 we have added a more detailed explanation to the discussion under the section “Foot placement control” (see manuscript for full section), as follows:

“The degree of foot placement control can be quantified as the relative explained variance (R2) of Model 1. A reduction in foot placement error could result in a higher R2, i.e. a higher degree of foot placement control. However, as illustrated in Fig. 13, apart from changes in foot placement error, other factors can influence the R2 as well. The changes in Fig. 13 panels C-D result in an increase in absolute explained variance. In earlier work, we referred to a higher absolute explained foot placement variance as an increase in the “foot placement contribution” (21).” 

(p.20, lines 421-427)

And:

“The degree of foot placement control (R2) takes both the absolute explained variance and the foot placement error into account, giving a good summary of foot placement control as a stability mechanism. However, only with an absolute measure of foot placement error can inferences be made regarding (improved) foot placement precision. A high absolute explained variance may lead to an overestimation of foot placement precision, when only considering the R2. In the other way around, a low absolute explained variance may lead to an underestimation of foot placement precision. Moreover, if the absolute explained variance and the foot placement error both increase/decrease, these changes may be hidden behind a constant R2, as is the case in the current study.” (p 20-21, lines 438-446)

The explanation was based on the model, but a physiological rationale was not provided. 

Physiologically speaking, we interpret the foot placement error as sensorimotor noise or loose control. I.e. participants do not succeed in placing their foot exactly in the location that they are aiming for based on the foot placement model due to noisy motor execution, and/or because of errors in their sensory estimation of the center-of-mass kinematic state. A reduction in the foot placement error could signify that participants learned to place their feet more precisely.

We have addressed this in the discussion under the section “Foot placement control” (see manuscript for full section) as well:

“…we interpret the foot placement control model (Model 1) as a feedback mechanism in which sensory information CoM kinematic state (12), is used to predict a target foot placement location (FP), that will ensure mediolateral gait stability (6). During swing, the swing leg is steered towards this target location (13, 18), however, once the foot is placed, actual foot placement does not exactly match the predicted/target foot placement location (FP) (7). 

The residuals of Model 1 denote these foot placement errors and can be considered as the result of sensorimotor noise or loose control. Decreasing these foot placement errors would mean more precise foot placement control, and this could be an aim for training.” (p.19 lines 411-420)

The explanation indicates that the lower magnitude of foot placement control may result from less error (i.e., less variance) in foot placement (lines 460-462). This argument seems problematic if it implies that large errors will result in large amount of control. Further clarification would be helpful. 

We aimed to clarify our reasoning here by adding Figure 13 panel C. If we place our feet correctly (I.e. with a lower error), the proper foot placement may result in lower CoM kinematic state variability. In response, in accordance with the model foot placement variance will decrease as well. Large errors would indeed lead to higher CoM variability, which requires more foot placement control. However, since in the latter case this is not combined with low foot placement errors, the degree of foot placement control would not be as high as when foot placement would be precise as well. So, again this is an example where it is important to consider two outcome measures related to the foot placement model.

We have already built up to lines 460-462 (now lines 515-521) by adding the section “Foot placement control” earlier in the discussion and have rewritten lines 515-521 in an attempt to better guide the reader through our rationale as follows:

 “Intuitively, one would expect that a decrease in foot placement error would coincide with a higher degree of foot placement control. However, going back to Fig. 13, we can see that this is not necessarily the case. Whereas a lower foot placement error can increase the degree of foot placement control, a lower CoM variance can decrease it, by lowering the absolute explained foot placement variance (i.e. the foot placement contribution). Given the unchanged degree of foot placement control, in combination with the reduction in foot placement error, we can infer that the foot placement contribution decreased.” (p. 24-25, lines 515-521)

Similarly, the observation that stability changes did not persist at retention, but foot placement error did persist appears inconsistent and also left me feeling dissatisfied.

This observation is indeed not in line with what we expected and leaves us speculating for an explanation. We expected the reduction in foot placement error to be reflected in a more stable walking pattern at retention, but as pointed out step-by-step foot placement control is not the only stability mechanism at play. Although foot placement control may have improved compared to the control week, other mechanisms may have been less used. Perhaps their improved foot placement control allowed our participants to loosen other stabilizing mechanisms that are energetically costlier. This was also the last training session, during which they may have felt the most comfortable. As such, they may have allowed themselves some slack concerning the use of other stability mechanisms. It can be noted however, that despite the non-significant effect, in Figure 12, visually, stability seems to be better during the retention test as compared to during the control week. 

We have decided to add some of the speculation as follows:

“The mismatch in the retention results of foot placement error and gait stability is unexpected. Improved foot placement error was expected to lead to improved gait stability. However, as mentioned above, none of our outcome measures exactly mirrored the changes in stability, suggesting that other stability control mechanisms play a role as well. We speculate that, although foot placement was tightly controlled (as reflected by the decreased foot placement error) during the retention test, other mechanisms could have been less used (36).” (p. 25-26, lines 542-547)

2. Please address the role of ML moments of the body besides the ankles. While the ankle ML moments were constrained, other ML moments such as trunk lateral flexors and the hips were not constrained. It seems they could take on (some of) the role of the ankle ML moments.

You make a good point that there may be other stability control mechanisms that compensate for the constrained ankle moments. Since we did not record full body kinematics nor EMG activity, we cannot do any further analysis on this based on the current data. We do agree it is good to put more emphasis on the possibility of an angular momentum strategy compensating for the ankle moment constraints. Therefore, we have now added the following to the discussion:

“It should be noted however, that even when participants fully complied with the instructions, we did not control the use of other stabilizing mechanisms, such as an angular momentum mechanism (9, 33), or modulation of stride frequency (34).” (p.22, lines 457-460)

3. While the result was statistically significant, please address the functional consequence of a 0.5 mm change in foot placement error. For example, can the authors estimate the corrective moment/torque needed to correct for a 0.5 mm foot placement error?

In general, we consider foot placement control as a center-of-pressure strategy, determining the moment of the ground reaction force that accelerates the center-of-mass in the mediolateral direction. As a simple example, we consider the vertical ground reaction force, and the center-of-mass as the pivot. The mediolateral distance between the projection of the center-of-mass on the floor and the center-of-pressure will then define the moment arm. 

We thus have: 

M = Fgrf_vert * (CoPml_axis – CoMml_axis)

Rounded up our participants had an average weight of 75 kg. So, let’s assume a vertical ground reaction force of 750N. They had step widths around 0.10 meter. For the sake of simplicity, let’s assume that the center-of-mass is positioned in between the two feet. If we take the stance foot as point 0, the average moment arm for our moment calculation is:

CoPml_axis – CoMml_axis = 0.1-0.05 = 0.05 m

And the average moment would be:

M = Fgrf_vert * (CoPml_axis – CoMml_axis) = 750 * 0.05 = 37.5 Nm

With a foot placement error of 0.5 mm too medial and the same Fgrf_vert we would compute:

M = Fgrf_vert * (CoPml_axis – CoMml_axis) = 750 * 0.0495 = 37.1250 Nm

So the difference in moment with a 0.5 mm foot placement error is 37.5-37.125 = 0.375 Nm.

The largest change we observed due to training was 2 mm in week 4. This would have led to a difference in moment of about:

M = Fgrf_vert * (CoPml_axis – CoMml_axis) = 750 * 0.048 = 36 Nm

37.5-36 = 1.5 Nm

Although the difference in torque does not seem very large, after foot placement corrections have to be made through ankle moment control or an angular momentum mechanism, which are the other two analytically distinguishable components contributing to the CoM acceleration [3, 4]. Since moments can be generated relatively energetically inexpensively through foot placement control (only the weight of the swing leg needs to be moved to prevent foot placement errors) [5], even such seemingly small reductions in foot placement error may be functionally beneficial. Apart from energy efficiency, during steady-state walking we can be at risk of falling if our center-of-mass moves close to the lateral border of the base of support. When near this border, even small corrections that help us to stay just at the medial side of this border may prevent a fall.

We have now added part of this calculation and reasoning to the discussion under the header “Functional relevance”, as follows:

“The largest training effect was observed during normal walking in week 4. In this week, foot placement error had reduced two millimeters compared to the control week. Foot placement is an effective stability mechanism as it largely determines the center-of-pressure possition, and consequently the resulting moment of the ground reaction force, which accelerates the CoM in the mediolateral direction [3, 4]. To estimate of a two millimeter foot placement error on the moment generated, we consider the vertical ground reaction force, and the center-of-mass as the pivot. The mediolateral distance between the projection of the center-of-mass on the floor and the center-of-presssure then defines the moment arm. We thus have:

M=F_grfvert*(CoP_MLaxis-CoM_(MLaxis))

In which M is the moment generated by the vertical ground reaction force, Fgrfvert is the vertical groun reaction force, CoPMLaxis is the mediolateral center-of-pressure position, and CoMMLaxis the mediolateral CoM position, both with respect to the other stance foot. 

Rounded up our participants had an average weight of 75 kilogram. So we will assume a vertical ground reaction force of 750 Newton. The participants walked with step widths around 0.10 meter. For the sake of simplicity, we assume that the center-of-mass is positioned in between the two feet. If this is the case, the average moment arm for our moment calculation is:

CoP_MLaxis- CoP_MLaxis=0.1-0.05=0.05 m 

And the average moment would be:

M=F_grfvert*(CoP_MLaxis-CoM_MLaxis )=750*0.05=37.5 Nm

A foot placement error of two millimeters too medial, and the same Fgrfvert, would computed:

M_error=F_grfvert*(CoP_MLaxis-CoM_MLaxis )=750*0.048=36 Nm

Thus, a corrective moment of 1.5 Nm would be needed to compensate for a foot placement error of two millimeters.

Although the difference between the two moment calculations (M and Merror) does not sseem very large, after erronous foot placement, a fast corrective response would call uupon an ankle or angular momentuum mechanism (i.e. the other two analytically distinguishable components contributing to the mediolateral CoM acceleration [3, 4]). During gait, stabilizing moments can be genereated relatively energetically inexpensively through foot placmeent control (only the weight of the swing leg needs to be moved to prevent foot placement errors) [5]. Therefore, even a two-millimeter reduction in foot placement error may be beneficial. Apart from energy efficiency, during steady-state walking, we can be at risk of falling sidewards if our center-of-mass moves close to the lateral border of our base of support. When near this border, even small corrections that help the CoM to stay just medially of this border may prevent a side-ward fall.” (p. 28-29, lines 598-636)

4. Consider adding medio-lateral to the title and abstract for clarity (i.e., ML ankle moments)

We have now added “Mediolateral” to the title: “Mediolateral foot placement can be trained: Older adults learn to walk more stable, when ankle moments are constrained”.

In addition we have added “mediolateral” to the following sentence of the abstract:

“…we attempted to train mediolateral foot placement control…” (p. 2, line 22) 

5. Some of the observed changes may result from requiring the participants to walk at the their preferred treadmill speed (determined with flat shoes) for all conditions. I expect their preferred gait speed on the training shoes would be slower if they had the option.

We deliberately took away the option of walking at a different gait speed, as changing gait speed could be a way to compensate for the ankle moment constraints, other than to improve the degree or precision of foot placement control. Moreover, by keeping the speed the same for all conditions, any changes in foot placement control can be attributed to constraining the ankle moment control, rather than to changes in gait speed [6]. 

We have now added a reason for choosing the same speed in all sessions and conditions to the methods:

“This walking speed was used in all subsequent conditions, sessions and weeks, to ensure that any changes in foot placement control could be attributed to constraining ankle moment control, rather than to changes in gait speed (25).” (p. 8 lines 167-169)

If they would have had the option to choose their preferred gait speed with LesSchuh, I would also not be sure whether they would choose a slower or a faster walking speed. From previous work, we know that when walking with LesSchuh, young adults tend to decrease their stride time [7]. Although not significant, the same trend is shown in the current study for older adults. 

6. The total duration of training was 90 minutes, which seems short relative to other training studies. Please comment, especially related to interpretations about strength enhancement (lines 469-470).

The duration of the training during each session was based on an earlier study in young adults [2]. Prior to the experiment, we considered that this would allow for insights into differences in (training) effects between young and older adults. However, as we figured out later, the use of a different treadmill [8], prevented us from making such comparisons.

The training (duration) was not designed to enhance strength. However, as mentioned in line 468, the subjective feedback from the participants prompted us to consider an explanation related to strength enhancement. However, we admit that in a follow-up study, a measure for strength enhancement is needed to provide further insights into the matter.

We have now admitted this in the manuscript by adding:

“… although we did not perform measurements to verify this. If so, the training…” (p. 25, lines 533-534)

7. Line 102 – correct the spelling of ‘anonymously’

Thank you for noticing this. We have corrected it now.

8. Text on lines 429+ discusses the differences in step width and stride time, but they were not significant. This text should be removed or it should be explicitly stated that these are trends when they are first discussed. It’s pointed out later, but currently the text reads as if the values are significantly different.

We have now added: 

“Although these trends were not significant…” 

to p. 23 lines 484-485 (previously lines 430-431) to avoid implying significance.

Reviewer #2: This paper makes a valuable contribution to research that tries to unravel (impaired) mediolateral balance control during walking. Although prior research from this group uses similar paradigms I want to emphasize the their study as a very nice and original example of experimental design. The study would have benefited from a control group, mainly to account for the effects of familiarization to treadmill walking. Nevertheless, this should not hold this work from being published in its current form (set apart some comments the authors can find below), as there are many insights to be gained form the manuscript as it is. The manuscript is very well structured.

1) Explain foot placement error and control by providing some examples.

An important comment that I would like to see addressed is on the explanation of foot placement error and foot placement control. These are the two most important concepts of the study and are not easy to comprehend. I like them as model of quantifying foot placement control. However, it took some time to understand exactly what these mean. I describe three scenarios below to check my understanding with the authors.

If these, or a variation of these, make sense, these could be added somewhere for clarification of this model? The wording can be improved and shortened probably.

• If FP varies a lot but COM motion is always the same, foot placement control is small and the foot placement error is large. Subjects are changing foot placement for no reason?

Indeed. Although we would say that given this statement foot placement is strongly affected by noise. This noise in part is inevitable, like any motor noise, but depending on the context participants could also choose to control their foot placement more or less, i.e. use more or less effort to counteract natural variability.

• If FP variance is large and can be explained by COM motion for 90%, subjects are adequately modulating foot placement to the needs of maintaining stability as COM is variable. Foot placement error is still large as 10% of the large original variance is high and might give balance problems or require an additional mechanism to control for balance.

• If FP variance is small and can be explained by COM motion for 90%, subjects are adequately adapting foot placement to the needs of maintaining stability as COM is variable. Foot placement error is small as 10% of original variance is low and should not give balance problems.

In general, an explained variance of 90% would be high, given what we measure from our participants. Having said that, we agree with your reasoning that a 10% error with larger FP variance would allow for more room for improvement than a 10% error with a smaller FP variance. Since the R2 would mask these differences, it is important to also consider foot placement error separately as an outcome measure.

I understand that these concepts might be explained in prior work of the authors. But I think the paper should stand on its own.

Thank you for this comment and sharing your thoughts on how you understand it. We think here indeed we got caught up in the pitfall of explaining things to ourselves, rather than to the reader. We have tried to illustrate the concepts in Figure 13. Moreover, we added a more detailed explanation in the Discussion under the section “Foot placement control” (see manuscript for full section), and tried to incorporate the points and statements you have made above:

“The degree of foot placement control (R2) takes both the absolute explained variance and the foot placement error into account, giving a good summary of foot placement control as a stability mechanism. However, only with an absolute measure of foot placement error can inferences be made regarding (improved) foot placement precision. A high absolute explained variance may lead to an overestimation of foot placement precision, when only considering the R2. In the other way around, a low absolute explained variance may lead to an underestimation of foot placement precision. Moreover, if the absolute explained variance and the foot placement error both increase/decrease, these changes may be hidden behind a constant R2, as is the case in the current study.”

(p. 20-21 lines 438-445)

2) Motivate the use of foot placement error instead of foot placement variability

When reading the results, I sometimes doubt whether it would be better to just look at foot placement variance instead of foot placement error. Can the authors defend their choice of choosing the placement error as they define it?

By looking at foot placement error, as the residual of a linear model correlating center-of-mass kinematic state with subsequent foot placement (i.e. Model 1 of this manuscript), we have a measure defining how precise we place our foot, given that we want to coordinate this foot placement with respect to the center-of-mass. When looking at foot placement variability alone, we cannot infer whether this variability is regulated to in accordance with variations in center-of-mass kinematic state. Since the coordination between foot placement and the center-of-mass kinematic state is important for stability control, choosing foot placement error as we define it, is in accordance with the aim of improving stability in older adults. We have motivated the use of foot placement error as follows: 

“we interpret the foot placement control model (Model 1) as a feedback mechanism in which sensory information CoM kinematic state (12), is used to predict a target foot placement location (FP), that will ensure mediolateral gait stability (6). During swing, the swing leg is steered towards this target location (13, 18), however, once the foot is placed, actual foot placement does not exactly match the predicted/target foot placement location (FP) (7). s

The residuals of Model 1 denote these foot placement errors and can be considered as the result of sensorimotor noise or loose control. Decreasing these foot placement errors would mean more precise foot placement control, and this could be an aim for training.” (p. 19, lines 411-420)

Admittedly, a combination of R2 alongside foot placement variance could provide similar information. However, we considered foot placement error to be more intuitive to the reader, as well as more directly related to the corrective mechanism LesSchuh is trying to constrain.

3) Can you bring the results together in one overview table/figure.

I think the existing figures can stay as they are. But, since so many results are to be interpreted it would be nice to have an overview table that summarizes what is changing and not changing across week/session. The reader can during reading the discussion than easily refer to this table.

We have now added a table at the end of the results section summarizing our main findings.

Some other minor comments

Figures: They are very low resolution in the manuscript we received. I assume this will be fixed.

That the resolution of the figures was not sufficient surprises us, since the uploaded files seem to have a good resolution on our own laptop. We will ask the journal how to deliver our files differently to achieve a better resolution.

L 102: Anonumously -- > anonymously

Thank you for noticing this. This has been corrected.

L442-446: Or gait stability as calculated here is a more subtle measure that can change for longer due to familiarization compared to foot placement control.

We agree that this could be an alternative explanation and have reflected a bit more on this in the manuscript as follows:

“It is likely that part of the improvement in gait stability can be attributed to the LesSchuh training, rather than just familiarization, since the first significant difference between weeks occurred after LesSchuh training. An alternative explanation that cannot be ruled out however, would be that treadmill familiarization occurs gradually across weeks, which for the local divergence exponent becomes apparent from week 3 onwards. Although several gait parameters have shown to stabilize after only six minutes of treadmill walking (30), it is still unclear after how much normal treadmill walking the local divergence exponent would demonstrate plateau values.” (p. 24, lines 499-506)

L528: Although the discussion explains all details well it seems that the conclusion might overemphasize the foot placement errors and gait stability. Maybe change to: … decreased foot placement errors and increased gait stability during normal walking may be explained by familiarization to treadmill walking and training with the LesSchuh. The extent to which they contribute requires a control study.

We think your suggestion is fair and have updated the conclusion as follows:

“We conclude that decreased foot placement errors and increased gait stability during normal walking may be explained by familiarization to treadmill walking and training with the LesSchuh. The extent to which familiarization and LesSchuh training contribute requires a control study.” (p. 30, lines 643-646)

Limitation: Can you mention the lack of a control group as a limitation that could have alleviated the confounding effects of familiarization?

We have now clearly mentioned this as a limitation of our study:

“The presence of a control group would have alleviated confounding effects of familiarization, and its absence is a clear limitation of this study. In future work, a randomized controlled trial, with normal treadmill walking as a control intervention could verify the presumed effects of LesSchuh.” (p. 27 lines 576-579)

Future work: The authors mention collecting full kinematics for these studies. This would be great for sure (control for and observe inertial compensation strategies). It is worth mentioning that this can be enabled by markerless mocap as I assume the choice to not collect full kinematics in this study was made because of time constraints.

Indeed, since we had quite a lot of sessions for the participants and we wanted to collect data of all of them, we compromised on the amount of markers we used. This also helped us to keep the experiments within a tolerable time for the participants. Thanks for suggesting markerless mocap for future experiments, this could definitely help in shortening the time the participants would need to be in the lab whilst still collecting full body kinematics!

Just a question out of interest: Is there a high risk of spraining your ankle with the LesSchuh?

So far in all of our studies we did not observe participants spraining their ankle whilst walking with LesSchuh. Based on this there doesn’t seem to be a high risk. The ridge underneath the shoe is quite low, which makes it safer for the participants. Whenever they fail to stay on the ridge, they basically “cheat” on the experiment and touch the treadmill with the shoe’s sole.

References

1. Wang, Y. and M. Srinivasan, Stepping in the direction of the fall: the next foot placement can be predicted from current upper body state in steady-state walking. Biology letters, 2014. 10(9).

2. Hoogstad, L.A., et al., Can foot placement during gait be trained? Adaptations in stability control when ankle moments are constrained. Journal of Biomechanics, 2022: p. 110990.

3. Hof, A.L., The equations of motion for a standing human reveal three mechanisms for balance. Journal of biomechanics, 2007. 40(2): p. 451-457.

4. van Leeuwen, M., S. Bruijn, and J. van Dieën, Mechanisms that stabilize human walking. Brazilian Journal of Motor Behavior, 2022. 16(5): p. 326-351.

5. Bruijn, S.M. and J.H. van Dieën, Control of human gait stability through foot placement. Journal of The Royal Society Interface, 2018. 15(143): p. 20170816.

6. Stimpson, K.H., et al., Effects of walking speed on the step-by-step control of step width. Journal of biomechanics, 2018. 68: p. 78-83.

7. van Leeuwen, A.M., et al., Active foot placement control ensures stable gait: Effect of constraints on foot placement and ankle moments. Plos one, 2020. 15(12): p. e0242215.

8. Hos, M., et al., Differential effects of ankle constraints on foot placement control between normal and split belt treadmills. Journal of Biomechanics, 2022: p. 111349.

---

## [Decision Letter · Decision Letter 1]

21 Sep 2023

Mediolateral foot placement control can be trained: Older adults learn to walk more stable, when ankle moments are constrained

PONE-D-23-09453R1

Dear Dr. van Leeuwen,

We’re pleased to inform you that your manuscript has been judged scientifically suitable for publication and will be formally accepted for publication once it meets all outstanding technical requirements.

Kind regards,

Ryan T. Roemmich

Academic Editor

PLOS ONE

Additional Editor Comments (optional):

Reviewers' comments:

Reviewer's Responses to Questions

**Comments to the Author**

1. If the authors have adequately addressed your comments raised in a previous round of review and you feel that this manuscript is now acceptable for publication, you may indicate that here to bypass the “Comments to the Author” section, enter your conflict of interest statement in the “Confidential to Editor” section, and submit your "Accept" recommendation.

Reviewer #1: All comments have been addressed

Reviewer #2: All comments have been addressed

2. Is the manuscript technically sound, and do the data support the conclusions?

Reviewer #1: Yes

Reviewer #2: Yes

3. Has the statistical analysis been performed appropriately and rigorously? 

Reviewer #1: Yes

Reviewer #2: Yes

4. Have the authors made all data underlying the findings in their manuscript fully available?

Reviewer #1: Yes

Reviewer #2: Yes

5. Is the manuscript presented in an intelligible fashion and written in standard English?

Reviewer #1: Yes

Reviewer #2: Yes

6. Review Comments to the Author

Reviewer #1: The authors have addressed all of my previous concerns. As I indicated in the first review, I find approach to training foot placement is innovative and creative. The manuscript is clear, and the study is relevant to both basic scientists and clinicians.

Reviewer #2: (No Response)

7. PLOS authors have the option to publish the peer review history of their article (what does this mean?). If published, this will include your full peer review and any attached files.

Reviewer #1: No

Reviewer #2: **Yes: **Tom Van Wouwe

---

## [Editor Report · Acceptance letter]

3 Oct 2023

PONE-D-23-09453R1 

Mediolateral foot placement control can be trained: Older adults learn to walk more stable, when ankle moments are constrained 

Dear Dr. van Leeuwen:

I'm pleased to inform you that your manuscript has been deemed suitable for publication in PLOS ONE. Congratulations! Your manuscript is now with our production department. 

Kind regards, 

on behalf of

Dr. Ryan T. Roemmich 

Academic Editor

PLOS ONE